# Debiased Causal Tree: Heterogeneous Treatment Effects Estimation with Unmeasured Confounding

**Caizhi Tang[1]\*** **Huiyuan Wang[2]\*** **Xinyu Li[2]** **Qing Cui[1]** **Ya-Lin Zhang[1]**

**Feng Zhu[1]** **Longfei Li[1]** **Jun Zhou[1]†** **Linbo Jiang[1]**

[1]Ant Group, [2]School of Mathematical Sciences, Peking University
{caizhi.tcz,cuiqing.cq,lyn.zyl,zhufeng.zhu
longyao.llf,jun.zhoujun,xiaobo.jlb}@antgroup.com
{huiyuan.wang,xinyu.li}@pku.edu.cn

## Abstract

Unmeasured confounding poses a significant threat to the validity of causal inference. Despite that various ad hoc methods are developed to remove confounding effects, they are subject to certain fairly strong assumptions. In this work, we consider the estimation of conditional causal effects in the presence of unmeasured confounding using observational data and historical controls. Under an interpretable transportability condition, we prove the partial identifiability of conditional average treatment effect on the treated group (CATT). For tree-based models, a new notion, *confounding entropy*, is proposed to measure the discrepancy introduced by unobserved confounders between the conditional outcome distribution of the treated and control groups. The confounding entropy generalizes conventional confounding bias, and can be estimated effectively using historical controls. We develop a new method, debiased causal tree, whose splitting rule is to minimize the empirical risk regularized by the confounding entropy. Notably, our method integrates current observational data (for empirical risk) and their historical controls (for confounding entropy) harmoniously. We highlight that, debiased causal tree can not only estimate CATT well in the presence of unmeasured confounding, but also is a robust estimator of conditional average treatment effect (CATE) against the imbalance of the treated and control populations when all confounders are observed. An extension of combining multiple debiased causal trees to further reduce biases by gradient boosting is considered. The computational feasibility and statistical power of our method are evidenced by simulations and a study of a credit card balance dataset.

## 1 Introduction

Conditional average treatment effect (CATE), also known as heterogeneous treatment effect, can characterize the causal effect of certain treatments on the outcome over the whole population as a function of measured covariates. Another closely related concept, conditional average treatment effect on the treated group (CATT), is more relevant when we focus on the effect of treatments for the sub-population which is actually exposed to certain specific treatments; for instance, we are more interested in the effect on children of taking anti-depressants during pregnancy for the people who

---

\*Equal contribution.
†Corresponding author.

36th Conference on Neural Information Processing Systems (NeurIPS 2022).

actually took the drug (Boukhris et al., 2016). The interpretable and informative nature has made it a great importance to estimate CATE and CATT from observational or experimental data in a wide spectrum of applications, including medical decision-making and personalized medicine (Glass et al., 2013; Obermeyer and Emanuel, 2016), evaluating advertising and socioeconomic policies (Breslow and Johnson, 1993; Athey, 2017; Gordon et al., 2019) and large-scale randomized experiments for recommendations (Li et al., 2010; Gilotte et al., 2018), etc.

In the absence of unmeasured confounding, there exist lots of methods developed to estimate CATE (CATT), including tree-based methods (Athey and Imbens, 2016; Wager and Athey, 2018; Athey et al., 2019), meta-learners (Künzel et al., 2019), R-learners (Nie and Wager, 2021), doubly robust learners (Kennedy, 2020), and so on. Basically, such an unconfoundedness condition hinges on an investigator's ability to accurately measure all related covariates, which is questionable in practice, especially when there exist covariate-dependent measurement errors (Wacholder, 1995; Tekwe et al., 2019), biased treatment allocations (Kalish and Begg, 1985; Tran and Zheleva, 2019) and batch effects (Gagnon-Bartsch and Speed, 2012; Goh et al., 2017). In the presence of unmeasured confounding, however, the identifiability of CATE or CATT cannot be ensured in general, leading to failures of the aforementioned estimation procedures.

To control the effect introduced by confounders, various methods have been developed, e.g., sensitivity analyses (Rosenbaum and Rubin, 1983), instrumental variable (IV) approaches (Angrist et al., 1996; Lin et al., 2015), and proximal causal learning (Kuroki and Pearl, 2014; Miao et al., 2018; Shi et al., 2020). Nonetheless, several limitations of these methods hinder their direct application in CATE/CATT estimation. For example, we cannot derive point estimation from sensitivity analyses, and even valid IVs can only partially identify CATE in principle (Swanson et al., 2018). Although ATE and ATT can be identified in proximal causal learning (Cui et al., 2020), the assumption that the measured covariates are correctly classified into three types of proxies requires prior knowledge.

Recently, a line of work proposed to combine randomized trials (Concato et al., 2000) and observational studies to adjust confounders, which typically requires transportability (Pearl and Bareinboim, 2011) of a randomized controlled trial to external observational studies (Rothwell, 2005). For example, Yang et al. (2020) proposed to use the confounding function which summarizes the impact of confounders to estimate heterogeneous treatment effects, where the treatment effect function is assumed to be transportable. For a comprehensive review, see Colnet et al. (2020). Problematically, many observational studies do not have related randomized trials due to practical and methodological issues such as cost and restrictive inclusion/exclusion criteria.

In this work, we move further in the direction of data fusion, without requiring experimental data. Instead, we consider the case where covariates and the outcome are collected at multiple timestamps, before and after the treatment. We take two timestamps $t \in \{t_1, t_2\}(t_1 < t_2)$, as an example, where the treatment is imposed at the time $t_2$. Following the potential outcome framework (Rubin, 1978), denote by $Y_t = DY_t^{(1)} + (1 - D)Y_t^{(0)}$ the observed outcomes at the time $t$, where $Y_t^{(d)}$ is the potential outcome for the treatment $D = d \in \{0, 1\}$. We remark that $Y_{t_1} = Y_{t_1}^{(0)}$ since the treatment does not take place at $t_1$. Let $X$ and $U$ be time-invariant covariates and unmeasured confounders, respectively. Our work is motivated by the following identity

$$E(Y_t^{(1)} \mid D = 1, X = x) - E(Y_t^{(0)} \mid D = 0, X = x) = \underbrace{E(Y_t^{(1)} - Y_t^{(0)} \mid D = 1, X = x)}_{\text{CATT}}$$
$$+ \underbrace{E(Y_t^{(0)} \mid D = 1, X = x) - E(Y_t^{(0)} \mid D = 0, X = x)}_{\text{Confounding Bias, } \boldsymbol{b}_t}, \qquad \forall t \tag{1}$$

which holds regardless of $U$. In general, the confounding bias (Bareinboim and Pearl, 2012; Yang et al., 2020) $\mathbf{b}_t$ does not vanish due to $U$. However, the smaller $\boldsymbol{b}_t$ is, the closer CATT could be to the difference in conditional expectations. We then propose a tree-based method, which, at high level, aims to (i) find an optimal partition of the feature space such that the confounding bias is smallest within each small region, and (ii) estimate the left hand of (1). Despite that the confounding bias is unavailable in general, we can represent it with tree-based models using data collected at $t = t_1$ since $Y_{t_1} = Y_{t_1}^{(0)}$ and it is known which individual would receive the treatment. If the optimal partition that leads to the smallest historical confounding bias changes "mildly" over time, i.e., it is transportable, the partition derived at time $t_1$ is also nearly optimal at $t_2$. The two steps can proceed simultaneously, and we calibrate and formalize the basic idea in Section 2.2.

Our contribution is summarized as follows: (i) we introduce a new notion, confounding entropy, to characterize the discrepancy between the conditional distributions of the treated and control groups that generalizes the confounding bias, and propose a tractable and scalable method, debiased causal tree, to estimate CATT from observational data before and after the treatment by developing a new criterion for node splitting, which minimizes both empirical risk and confounding entropy at the same time; (ii) to further reduce the bias, we integrate multiple causal trees by gradient boosting, which we refer to as Gradient Boosting Causal Tree (GBCT); (iii) under the unconfoundedness condition, we show that GBCT is robust to imbalance of the treated and control populations and outperforms other compared methods, which is obvious in severely imbalanced cases.

## 2   Methodology

We first introduce the problem setup, related notation, and basic ideas behind our proposed method.

### 2.1   Problem Setup

To begin with, we consider the case of binary treatment where $D = d \in \{0, 1\}$ denotes the treatment status. Let $Y_t^{(d)} \in \mathbb{R}$ denote the potential outcome at the time $t \in \{t_1, \ldots, t_m\}$ for the treatment $D = d$, $X \in [0, 1]^p$ denote a time-invariant $p$-dimensional covariate and $U$ denote a possibly time-varying unmeasured confounder vector. We maintain the classic SUTVA (Rubin, 1980) assumption that no interference between units and no hidden variations of treatments occur. We impose the treatment at the time $t = t_m$, and the observed outcome $Y_{t_m} = DY_{t_m}^{(1)} + (1 - D)Y_{t_m}^{(0)}$; before the treatment occurs, the historical outcomes $Y_{t_k} = Y_{t_k}^{(0)}, 1 \leq k \leq m - 1$. Here, we further impose the following condition.

**Assumption 1.** *For $d \in \{0, 1\}$, $P(D = d \mid X) > 0$ and $Y_{t_m}^{(d)} \perp\!\!\!\perp D \mid (X, U)$.*

Given $n$ triplets of i.i.d. samples $\{(X_i, \{Y_{i,t_k}\}_{k=1}^m, D_i)\}_{i=1}^n$, our goal is to estimate the CATT $\eta^*(x) \equiv E(Y_{t_m}^{(1)} - Y_{t_m}^{(0)} \mid D = 1, X = x)$. In general, $\eta^*$ is not identifiable due to unmeasured confounding; however we can prove that it is partially identifiable under certain transportability condition and Assumption 1. In the following, we denote the indicator function of event $A$ by I$(A)$.

Equation (1) implies that, $\eta^*(x) \approx \delta_{t_m}(x) \equiv E(Y_{t_m}^{(1)} \mid D = 1, X = x) - E(Y_{t_m}^{(0)} \mid D = 0, X = x)$ if the confounding bias $\boldsymbol{b}_{t_m}(x) \equiv E(Y_{t_m}^{(0)} \mid D = 1, X = x) - E(Y_{t_m}^{(0)} \mid D = 0, X = x) \approx 0$. In general, $\left|E_X[\boldsymbol{b}_{t_m}(X)]\right| \geq c_0 > 0$ due to unmeasured confounding. Proper stratification (Kernan, 1999), notably, can help mitigate confounding effects by introducing multiple subgroups within which confounders remain almost constant. By exploiting historical controls that contain information of unmeasured confounding, we propose to find the optimal partition of the feature space, e.g., $\{Q_j\}_{j=1}^q$ with $\cup_{j=1}^q Q_j = [0, 1]^p$, such that $\boldsymbol{b}_{t_m}(Q_j) \equiv E(Y_{t_m}^{(0)} \mid D = 1, X \in Q_j) - E(Y_{t_m}^{(0)} \mid D = 0, X \in Q_j) \approx 0$, for each $j = 1, \ldots, q$; or alternatively,

$$(Q_1^*, \ldots, Q_q^*) = \operatorname*{argmin}_{Q_j : 1 \leq j \leq q} \sum_{j=1}^q \boldsymbol{b}_{t_m}^2(Q_j). \tag{2}$$

Once $(Q_1^*, \ldots, Q_q^*)$ being derived, by (1), we can choose

$$\eta(x) = \sum_{j=1}^q \delta_{t_m}(Q_j^*) \mathrm{I}\{x \in Q_j^*\} \tag{3}$$

to approximate $\eta^*(\cdot)$, where $\delta_{t_m}(Q_j^*) = E(Y_{t_m}^{(1)} \mid D = 1, X \in Q_j^*) - E(Y_{t_m}^{(0)} \mid D = 0, X \in Q_j^*)$ is a constant function that approximates $\delta_{t_m}(x)$ for $x \in Q_j^*$. In (3), choosing a constant function to approximate $\delta_{t_m}(x)$ within $Q_j^*$ is only for simplicity, and local polynomials of any degree are applicable. As expected, $\eta(x)$ is close to $\eta^*(x)$ provided that (2) implies $\max_{1 \leq j \leq q} \boldsymbol{b}_{t_m}^2(Q_j^*) \approx 0$. A major difficulty of solving (2) is the unavailability of $\boldsymbol{b}_{t_m}(Q_j)$, $1 \leq j \leq q$, since $Y_{t_m}^{(0)}$ is a counterfactual for $D = 1$. Fortunately, since $D_i$ is known for each $i$ and all units stay untreated

before $t_m$, $\boldsymbol{b}_t(Q_j)$ for all $t \in \{t_1, \ldots, t_{m-1}\}$ are identifiable. Instead of (2), consider

$$(Q_1^{**}, \ldots, Q_q^{**}) = \underset{Q_j : 1 \leq j \leq q}{\operatorname{argmin}} \frac{1}{m-1} \sum_{k=1}^{m-1} \sum_{j=1}^{q} \boldsymbol{b}_{t_k}^2(Q_j). \tag{4}$$

The partition $\boldsymbol{Q}^{**} = \{Q_j^{**}\}_{j=1}^q$ of the feature space yields minimal historical confounding bias. If confounding effects are independent of time, $Q_j^{**} = Q_j^*$ holds for each $j = 1, \ldots, q$. We impose the following condition to ensure transportability from the historical optimal partition to its current counterpart.

**Assumption 2.** *For the partition $\{Q_j^{**}\}_{j=1}^q$ defined in (4), $B(t) \equiv \log \left\{ \sum_{j=1}^q \boldsymbol{b}_t^2(Q_j^{**}) \right\} : t \mapsto B(t) \in \mathbb{R}$ is a Lipschitz continuous function with parameter $L$; that is,*

$$\left| B(\tau_1) - B(\tau_2) \right| \leq L |\tau_1 - \tau_2|, \quad \text{for any } \tau_1, \tau_2 \geq 0.$$

Assumption 2 is testable empirically for $t \in [0, t_m)$, provided that historical controls are collected at sufficiently many timestamps. Under Assumption 2, we prove in the Appendix that $Q_j^* \approx Q_j^{**}$ *in effect*; that is, the solution to (4) is transferrable or transportable to the current data source. The partial identifiability of $\eta^*(x)$ is justified by the following theorem.

**Theorem 1.** *Under Assumption 2, for the partition $\{Q_j^{**}\}_{j=1}^q$ of $[0,1]^p$ defined in (4), it holds that*

$$\sup_{x \in [0,1]^p} \left| \eta(x) - \eta^*(x) \right| \leq \left\{ \frac{C_L}{m-1} \sum_{k=1}^{m-1} \sum_{j=1}^{q} \boldsymbol{b}_{t_k}^2(Q_j^{**}) \right\}^{1/2} + \max_{1 \leq j \leq q} \sup_{x \in Q_j^{**}} \left| \eta^*(x) - \eta^*(Q_j^{**}) \right|, \tag{5}$$

*where $\eta^*(Q_j^{**}) = E\left( Y_{t_m}^{(1)} - Y_{t_m}^{(0)} \mid D = 1, X \in Q_j^{**} \right)$ and $C_L$ depends only on $L$ and $t_m - t_1$.*

The first term in (5) denotes the confounding bias, and hinges on the transportability assumed in Assumption 2, while the second term measures the error of using a piecewise constant function to approximate $\eta^*(\cdot)$. Theorem 1 shows that $\eta^*(\cdot)$ is identifiable up to a band; this property that is referred to as partial identification (Swanson et al., 2018). Interestingly, the width of band in our case can be close enough to zero under stronger assumptions. For example, the confounding bias can be small if the stratification $\{Q_j^{**}\}_{j=1}^q$ can effectively adjust for confounding, and under certain smoothness condition, the approximation error decays to zero rapidly as the maximal volume of $Q_j^{**}$ tends to zero. *Our goal is to develop a computational feasible method to solve (4) and estimate $\eta^*(x)$ as in (3) substituting $Q_j^*$ with $Q_j^{**}$, for each $j$.*

## 2.2 Debiased Causal Tree

When it comes to stratifying or segmenting the feature space, the perhaps mostly used algorithms are tree-based methods (James et al., 2021), including classification and regression tree (CART) (Breiman et al., 1984), random forest (Breiman, 1996), and gradient boosting trees (Friedman, 2001). A regression tree $T(x; \boldsymbol{Q}, \boldsymbol{\mu})$ consists of two components: a set of leafs that partition the feature space, and the corresponding parameter $\boldsymbol{\mu} = \boldsymbol{\mu}(\boldsymbol{Q}) = (\mu(Q_j))$, where $Q_j$ is the region partitioned by the $j$-th leaf, $1 \leq j \leq q$, and $q$ is the total number of leafs. Typically, given a partition $\boldsymbol{Q}$ and $n$ pairs of i.i.d samples $\{(X_i, y_i)\}_{i=1}^n$, the optimal data driven estimator is $T(x; \boldsymbol{Q}, \widehat{\boldsymbol{\mu}}) = \sum_{j=1}^q \mathrm{I}\{x \in Q_j\} \widehat{\mu}(Q_j)$, where

$$\widehat{\mu}(Q_j) = (|\{i : X_i \in Q_j\}|)^{-1} \sum_{i=1}^{n} y_i \, \mathrm{I}\{X_i \in Q_j\}. \tag{6}$$

We define the auxiliary trees $T(x; \boldsymbol{Q}, \boldsymbol{\theta}_t^{(d)})$ to represent confounding biases, where $\boldsymbol{\theta}_t^{(d)} = (\theta_{t,j}^{(d)})$ such that $\theta_{t,j}^{(d)} = \theta_t^{(d)}(Q_j)$ fits the conditional mean $E(Y_t^{(0)} \mid D = d, X \in Q_j)$ for $d = 0, 1, t < t_m$, and define the main tree $T(x; \boldsymbol{Q}, \boldsymbol{\mu}^{(d)})$ to learn CATT, where $\boldsymbol{\mu}^{(d)} = (\mu_j^{(d)})$ such that $\mu_j^{(d)} = \mu^{(d)}(Q_j)$ fits the conditional mean $E(Y_{t_m}^{(d)} \mid D = d, X \in Q_j)$ for $d = 0, 1$. Auxiliary and main trees share the same splitting rules.

**Confounding entropy.** Treating the partition $\{Q_j\}_{j=1}^q$ as a parameter to be optimized, by (6), we estimate $\boldsymbol{b}_t(Q_j)$ for each $t \in \{t_1, \ldots, t_{m-1}\}$ by

$$\widehat{\boldsymbol{b}}_t(Q_j) = \widehat{\theta}_t^{(1)}(Q_j) - \widehat{\theta}_t^{(0)}(Q_j), \quad \text{where} \quad \widehat{\theta}_t^{(d)}(Q_j) = \sum_{i=1}^n \frac{\mathrm{I}\{X_i \in Q_j, D_i = d\}Y_{i,t}^{(0)}}{|\{i : X_i \in Q_j, D_i = d\}|}. \quad (7)$$

For simplicity, we write $\widehat{\theta}_t^{(d)}(Q_j)$ as $\widehat{\theta}_{t,j}^{(d)}$. Plugging $\widehat{\boldsymbol{b}}_t(Q_j)$ in (7) into (4), we obtain the estimator of the optimal partitions. Nonetheless, $\widehat{\boldsymbol{b}}_t(Q_j)$ as well as its population counterpart $\boldsymbol{b}_t(Q_j)$ only summarizes information of the first-order moment, and thus can give rise to partitions that yield estimators with large variances. This motivates us to model the discrepancy between $P_{t,1,j} = P(Y_t^{(0)} \mid D = 1, X \in Q_j)$ and $P_{t,0,j} = P(Y_t^{(0)} \mid D = 0, X \in Q_j)$ using symmetric cross entropy (Wang et al., 2019), defined as $\mathrm{H}(P_{t,1,j}, P_{t,0,j}) = -E_{P_{t,1}}[\log(P_{t,0,j})] - E_{P_{t,0,j}}[\log(P_{t,1,j})]$, which we refer to as the *confounding entropy* for time $t$ and region $Q_j$. For each $t < t_m$, parametrize the working densities of $P_{t,1,j}$ and $P_{t,0,j}$ by $p(y_t; \theta_{t,j}^{*(1)})$ and $p(y_t; \theta_{t,j}^{*(0)})$, respectively. In our case, the parameter is the conditional mean $\theta_{t,j}^{*(d)} = E(Y_t^{(0)} \mid D = d, X \in Q_j)$ for both $d = 0, 1$. The estimator of symmetric cross entropy between $P_{t,1,j}$ and $P_{t,0,j}$ can be obtained by cross-fitting the negative log-likelihood; that is,

$$\widehat{\mathrm{H}}_t(Q_j) \equiv - \sum_{i:X_i \in Q_j} \left\{ \frac{D_i \log(p(Y_{i,t}; \widehat{\theta}_{t,j}^{(0)}))}{|\{i : X_i \in Q_j, D_i = 1\}|} + \frac{(1 - D_i) \log(p(Y_{i,t}; \widehat{\theta}_{t,j}^{(1)}))}{|\{i : X_i \in Q_j, D_i = 0\}|} \right\}. \quad (8)$$

Note that for two distributions $P$ and $Q$ with density functions $p$ and $q$ respectively,

$$\mathrm{H}(P,Q) = -E_P\{\log p(X)\} + -E_Q\{\log q(X)\} + E_P\left\{ \frac{q(X)}{p(X)} \log \frac{q(X)}{p(X)} \right\} + E_Q\left\{ \frac{p(X)}{q(X)} \log \frac{p(X)}{q(X)} \right\}$$

$$= \mathrm{Entropy}(P) + \mathrm{Entropy}(Q) + \mathrm{KL}(Q,P) + \mathrm{KL}(P,Q).$$

Thus, confounding entropy can not only measure discrepancy between distributions, but also control the uncertainty/complexity, which avoids overfitting. As a concrete example, for isotropic Gaussian distributions, $\log(p(y_t; \theta_{t,j}^{*(d)})) \propto -(y_t - \theta_{t,j}^{*(d)})^2, d = 0, 1$,

$$\widehat{\mathrm{H}}_t(Q_j) = 2(\widehat{\theta}_{t,j}^{(0)} - \widehat{\theta}_{t,j}^{(1)})^2 + \sum_{i:X_i \in Q_j} \left\{ \frac{D_i(Y_{i,t} - \widehat{\theta}_{t,j}^{(1)})^2}{|\{i : X_i \in Q_j, D_i = 1\}|} + \frac{(1 - D_i)(Y_{i,t} - \widehat{\theta}_{t,j}^{(0)})^2}{|\{i : X_i \in Q_j, D_i = 0\}|} \right\}, \quad (9)$$

where, the first term is exactly $\widehat{b}_t^2(Q_j)$ defined in (7), and the second term is the sum of sample variances of $\widehat{\theta}_{t,j}^{(0)}$ and $\widehat{\theta}_{t,j}^{(1)}$. Therefore, both confounding bias and variance can be emphasized. In the end, assuming historical data collected at distinct timestamps are independent, the confounding entropy of all historical controls $\mathrm{H}(\Pi_{t<t_m}\Pi_{j=1}^q P_{t,1,j}, \Pi_{t<t_m}\Pi_{j=1}^q P_{t,0,j})$ is estimated by

$$\widehat{\mathrm{H}}(\boldsymbol{Q}) \equiv \frac{1}{m-1} \sum_{k=1}^{m-1} \sum_{j=1}^q \widehat{\mathrm{H}}_{t_k}(Q_j), \quad \widehat{\mathrm{H}}_{t_k}(Q_j) \text{ is defined in (9)}. \quad (10)$$

We remark that (10) can be interpreted as a marginally independent composite likelihood estimator (Varin et al., 2011), which can be readily generalized to model potential temporal correlation among $\{Y_{t_k,i}\}_{k,i=1}^{(m-1),n}$. However, for simplicity, we presume no temporal dependencies.

**Estimating CATT.** On the basis of (6), we then can estimate $\eta^*(x)$ with

$$\widehat{\eta}_h(x) = T(x; \widehat{\boldsymbol{Q}}_h, \widehat{\boldsymbol{\mu}}^{(1)}(\widehat{\boldsymbol{Q}}_h))) - T(x; \widehat{\boldsymbol{Q}}_h, \widehat{\boldsymbol{\mu}}^{(0)}(\widehat{\boldsymbol{Q}}_h)), \quad \widehat{\boldsymbol{\mu}}^{(d)}(\boldsymbol{Q}) = (\widehat{\mu}^{(d)}(Q_j)), \quad (11)$$

where by (6), $\widehat{\mu}^{(d)}(Q_j) = \sum_{i:X_i \in Q_j}(Y_{i,t_m}^{(d)}\mathrm{I}\{D_i = d\})/|\{i : X_i \in Q_j, D_i = d\}|, d = 0, 1$ for any given $\boldsymbol{Q}$, and $\widehat{\boldsymbol{Q}}_h = \{\widehat{Q}_j^h\}_{j=1}^q$ denotes the minimizer of $\widehat{\mathrm{H}}(\boldsymbol{Q})$ in (10). Here, the superscript "h" denotes the fact that only historical data are used to learn $\boldsymbol{Q}^{**}$. This two-stage estimator $\widehat{\eta}_h(x)$ may suffer from potential heterogeneity between historical and current data sources, since splitting mechanism of $\widehat{\eta}_h(x)$ is independent of current data set $\{(X_i, Y_{i,t_m}, D_i)\}_{i=1}^n$. It relies crucially on the transportability of historical optimal partitions $\{Q_j^{**}\}_{j=1}^q$. To leverage information from the current data, we also use trees $T(x; \boldsymbol{Q}, \widehat{\boldsymbol{\mu}}^{(d)}(\boldsymbol{Q}))$ to learn $E(Y_{t_m}^{(d)} \mid D = d, X = x)$. Let $\widehat{m}(x, D; \boldsymbol{Q}) = DT(x; \boldsymbol{Q}, \widehat{\boldsymbol{\mu}}^{(1)}(\boldsymbol{Q})) + (1 - D)T(x; \boldsymbol{Q}, \widehat{\boldsymbol{\mu}}^{(0)}(\boldsymbol{Q}))$. We aim to find a $\boldsymbol{Q}$ such that $\widehat{m}(x, D; \boldsymbol{Q})$ can predict $Y_{t_m}$ well meanwhile $\widehat{\mathrm{H}}(\boldsymbol{Q})$ being small.

**Regularized Regression Tree Estimator.** For any problem-specific loss function $\ell(\cdot, \cdot)$, e.g., squared loss for regression and cross-entropy for classification, we propose the following regularized empirical risk minimization:

$$\widehat{\boldsymbol{Q}} = \underset{\boldsymbol{Q}}{\operatorname{argmin}} \left\{ \frac{1}{n} \sum_{i=1}^{n} \ell\big(\widehat{m}(X_i, D_i; \boldsymbol{Q}), Y_{i,t_m}\big) + \lambda \widehat{\mathrm{H}}(\boldsymbol{Q}) \right\}, \tag{12}$$

where $\lambda$ denotes the tuning parameter, and the CATT estimator is given by $\widehat{\eta}(x) = T(x; \widehat{\boldsymbol{Q}}, \widehat{\boldsymbol{\mu}}^{(1)}(\widehat{\boldsymbol{Q}})) - T(x; \widehat{\boldsymbol{Q}}, \widehat{\boldsymbol{\mu}}^{(0)}(\widehat{\boldsymbol{Q}}))$.

We remark that there exists a trade-off between data fidelity of the current source and transportability of historical sources for the choice of $\lambda$. When $\lambda \to \infty$, $\widehat{\eta}(\cdot)$ is reduced to the naive estimator, ignoring the information of current data. In contrast, $\lambda = 0$ means that we only use current data to learn CATT, which, under the unconfoundedness condition, is similar in spirit to causal tree estimators proposed in Wager and Athey (2018) and Athey and Imbens (2016). For the optimal choice of $\lambda$, our method minimizes the confounding bias introduced by unmeasured confounders that deteriorate the performance of conventional causal trees, and thus the name "debiased causal tree" follows.

## 2.3 Ensemble of Debiased Causal Trees by Gradient Boosting

Ensemble is a commonly used method to improve the performance of tree-based models. Boosting and bagging are two representative methods, where the former is designed to reduce bias, and the latter focuses on variance reduction. Existing tree-based methods developed for treatment effect estimation combine multiple trees to make a random forest, since unregularized decision trees are prone to overfitting, especially when the tree in employment is deep. In our case, the objective function (12) includes a regularization term. As a consequence, our estimator tends to be underfitting instead of overfitting, particularly when the working densities that define (8) are misspecified and/or the tuning parameter $\lambda$ is relatively large. This motivates us to choose the boosting framework to improve the performance of single debiased causal tree. We define the boosting of $k$ trees as $F(x; \boldsymbol{\mathcal{Q}}_k, M_k) = \sum_{i=1}^{k} T(x; \boldsymbol{Q}_i, \boldsymbol{\mu}_i)$, where $\boldsymbol{\mathcal{Q}}_k = \{\boldsymbol{Q}_1, \ldots, \boldsymbol{Q}_k\} \equiv \boldsymbol{\mathcal{Q}}_{k-1} \cup \{\boldsymbol{Q}_k\}$ denotes the set of partitions of each tree, and $M_k = \{\boldsymbol{\mu}_1, \ldots, \boldsymbol{\mu}_k\} \equiv M_{k-1} \cup \{\boldsymbol{\mu}_k\}$ denotes the set of corresponding parameters. To highlight the difference, we use $\boldsymbol{\Theta}_k = \{\boldsymbol{\theta}_1, \ldots, \boldsymbol{\theta}_k\}$ to denote parameters of auxiliary tree models that measure confounding entropy.

At the $k$-th iteration, we have already obtained $k-1$ trees, $F^{(d)}(x; \widehat{\boldsymbol{\mathcal{Q}}}_{k-1}, \widehat{\boldsymbol{M}}_{k-1}^{(d)})$, to learn $E(Y_{t_m}^{(d)} \mid D = d, X = x), d = 0, 1$, whose difference yields CATT, and $k-1$ trees, $F_{\mathrm{bias}}^{(d)}(x; \widehat{\boldsymbol{\mathcal{Q}}}_{k-1}, \widehat{\boldsymbol{\Theta}}_{k-1}^{(d)})$, to learn $E(Y_t \mid D = d, X = x), t < t_m$, whose difference yields confounding bias. Notice that the trees for confounding bias share the same splitting regions with trees for CATT. In the following, we illustrate how to obtain the $k$-th tree.

Parallel to a single debiased causal tree, we first show a refined representation of confounding entropy by integrating multiple trees. Following the gradient boosting algorithm (Friedman, 2001), the $k$-th tree $T(x; \boldsymbol{Q}_k, \boldsymbol{\theta}_k^{(d)})$ manages to fit the residual, $Y_{i,t}(X_i) - F_{\mathrm{bias}}^{(d)}(X_i; \widehat{\boldsymbol{\mathcal{Q}}}_{k-1}, \widehat{\boldsymbol{\Theta}}_{k-1}^{(d)})$ with $D_i = d$ for each $d = 0, 1$. Denote by $\widehat{\boldsymbol{\theta}}_k^{(d)}(\boldsymbol{Q}_k) = (\widehat{\theta}_k^{(d)}(Q_{k,j}))$ the local mean for any partition $\boldsymbol{Q}_k$, where $\widehat{\theta}_k^{(d)}(Q_{k,j})$ is defined similarly to (7) with $Y_{i,t}$ being replaced by residuals. Similar to (8), we use the cross mean squared error to represent the confounding entropy

$$\widehat{\mathrm{H}}_t(\boldsymbol{Q}_k) \equiv \sum_{i=1}^{n} \left\{ \frac{(1 - D_i)\big(Y_{i,t} - F_{\mathrm{bias}}^{(1)}(X_i; \widetilde{\boldsymbol{\mathcal{Q}}}_k, \widehat{\boldsymbol{\Theta}}_k^{(1)})\big)^2}{|\{i : D_i = 0\}|} + \frac{D_i\big(Y_{i,t} - F_{\mathrm{bias}}^{(0)}(X_i; \widetilde{\boldsymbol{\mathcal{Q}}}_k, \widehat{\boldsymbol{\Theta}}_k^{(0)})\big)^2}{|\{i : D_i = 1\}|} \right\}, \tag{13}$$

where $\widetilde{\boldsymbol{\mathcal{Q}}}_k = \widehat{\boldsymbol{\mathcal{Q}}}_{k-1} \cup \{\boldsymbol{Q}_k\}$, $\widehat{\boldsymbol{\Theta}}_k^{(d)} = \widehat{\boldsymbol{\Theta}}_{k-1}^{(d)} \cup \{\widehat{\boldsymbol{\theta}}_k^{(d)}(\widehat{\boldsymbol{Q}}_k)\}$, and $Y_{i,t}^{(d)}$ denotes $Y_{i,t}$ with $D_i = d$.

It suffices to consider the integration of trees to learn $\delta_{t_m}(x)$. Again, the $k$-th tree $T(x; \boldsymbol{Q}_k, \boldsymbol{\mu}_k^{(d)})$ manages to fit the residual, $\widehat{R}_i^{(d)} = Y_{i,t_m}^{(d)} - F^{(d)}(X_i; \widehat{\boldsymbol{\mathcal{Q}}}_{k-1}, \widehat{\boldsymbol{M}}_{k-1}^{(d)})$ with $D_i = d$. Let $\widehat{\boldsymbol{\mu}}_k^{(d)}(\boldsymbol{Q}_k) = (\widehat{\mu}_k^{(d)}(Q_{k,j}))$ be the local mean for any partition $\boldsymbol{Q}_k$, where $\widehat{\mu}_k^{(d)}(Q_{k,j})$ is defined similarly to (11)

with $Y_{i,t_m}^{(d)}$ being replaced by $\widehat{R}_i^{(d)}$. The objective function for the $k$-th tree is displayed as follows:

$$\widehat{\boldsymbol{Q}}_k = \underset{\boldsymbol{Q}_k}{\operatorname{argmin}} \left\{ \frac{1}{n} \sum_{i=1}^{n} \ell(\widehat{R}_i^{(D_i)}, T(X_i; \boldsymbol{Q}_k, \widehat{\boldsymbol{\mu}}_k^{(D_i)}(\boldsymbol{Q}_k))) + \frac{\lambda}{m-1} \sum_{u=1}^{m-1} \widehat{\mathrm{H}}_{t_u}(\boldsymbol{Q}_k) \right\}. \tag{14}$$

Update $\widehat{\boldsymbol{\mathcal{Q}}}_k = \widehat{\boldsymbol{\mathcal{Q}}}_{k-1} \cup \{\widehat{\boldsymbol{Q}}_k\}$, and $\widehat{\boldsymbol{M}}_k^{(d)} = \widehat{\boldsymbol{M}}_{k-1}^{(d)} \cup \{\widehat{\boldsymbol{\mu}}_k^{(d)}(\widehat{\boldsymbol{Q}}_k)\}, d = 0, 1$. The CATT estimator at the $k$-th iteration is $\widehat{\eta}(x) = F^{(1)}(x; \widehat{\boldsymbol{\mathcal{Q}}}_k, \widehat{\boldsymbol{M}}_k^{(1)}) - F^{(0)}(x; \widehat{\boldsymbol{\mathcal{Q}}}_k, \widehat{\boldsymbol{M}}_k^{(0)})$.

We refer to our proposed method as Gradient Boosting Causal Tree (GBCT). We remark that the objective function (14) is greedily minimized by gradually adding the tree. Also, to speed up the computation and stabilize the algorithm, we follow XGBoost to approximate the objective function with a second-order Taylor expansion and exploit a squared $\ell_2$ regularization. See Chen and Guestrin (2016) for details.

## 3 Applicability of Debiased Causal Trees Without Confounders

In this section, we discuss the applicability of debiased causal trees without confounders. Despite that the confounding bias $\boldsymbol{b}_t(x) \equiv 0$ for any $t$ and $x$ without unmeasured confounding, numerically, the term $\widehat{\boldsymbol{b}}_t(Q_j)$ in (7) may not vanish, especially when the minimal sample size is small. This issue is substantive when treated and untreated units are imbalanced. Notably, the imbalance can significantly degrade the performance of T-learners and S-learners (e.g., see Künzel et al., 2019). Given the historical training set $\{(X_i, Y_{i,t}, D_i)\}_{i=1}^n$, the smaller empirical confounding entropy is for the partition $\boldsymbol{Q}$, the more balanced samples can be within each small region. Here, for simplicity, we consider $m = 2$ such that only one historical data source is used. Let $n_d(Q_j) = |\{i : D_i = d, X_i \in Q_j\}|$ be the number of samples in $Q_j$ with $D_i = d$. We formalize the above intuition with the empirical confounding bias $\widehat{\mathrm{H}}(\boldsymbol{Q})$ defined in (10) with working distribution being Gaussian.

**Proposition 1.** *Suppose there are no unobserved confounders, and if $Y_{i,t} \sim Y_t$ with mean $\mu_t$ and variance $\sigma_t^2$ independently for a given $t < t_m$, then, under the Gaussian specification of working densities, it holds that $E\left[\widehat{\mathrm{H}}(\boldsymbol{Q})\right] = 2\sigma_t^2 + \sigma_t^2 \sum_{j=1}^q (n_0(Q_j) + n_1(Q_j))/(n_0(Q_j)n_1(Q_j))$.*

By Proposition 1 and the harmonic mean inequality, to reduce $\widehat{\mathrm{H}}(\boldsymbol{Q})$, it suffices to find a $Q_j$ such that $n_0(Q_j) \approx n_1(Q_j)$, especially for $Q_j$ such that $\max\{n_0(Q_j), n_1(Q_j)\}$ is small. Numerical analysis can be found in Figure 1 in the Appendix, which indicates that, our proposed debiased causal tree is robust to imbalance of treatments.

## 4 Experiments

We conduct experiments with both simulated and real data for a better understanding of our proposed strategy. We investigate two scenarios in simulation studies: one with and one without unobserved confounding. When implementing GBCT, we subtract the empirical pre-treatment confounding bias from effect estimates as a trick to improve performance, i.e., $\widehat{\eta}(x) = F^{(1)}(x; \widehat{\boldsymbol{\mathcal{Q}}}_k, \widehat{\boldsymbol{M}}_k^{(1)}) - F^{(0)}(x; \widehat{\boldsymbol{\mathcal{Q}}}_k, \widehat{\boldsymbol{M}}_k^{(0)}) - \frac{1}{m-1} \sum_{u=1}^{m-1} \sum_{Q \in \widehat{\boldsymbol{\mathcal{Q}}}_k} \widehat{\boldsymbol{b}}_{t_u}(Q)$. We evaluate GBCT against state-of-the-art causal inference algorithms: (i) meta learners (Künzel et al., 2019), including TLearner, SLearner and Xlearner; (ii) causal forests, which are forest-based methods to model the treatment effects, including generalized causal forest (GRF, Athey et al., 2019), doubly robust forest (DR-RF, Bang and Robins, 2005) and double machine learning forest (DML-RF, Chernozhukov et al., 2018). These benchmarks are implemented by the third-party library ECOML (Keith et al., 2019).

For benchmark methods, both covariates $X$ and pre-treatment outcomes are used as input features. In contrast, our proposed GBCT employs an alternative way of exploiting historical controls, i.e., building models on $X$ while treating pre-treatment outcomes as labels to de-bias via the confounding entropy. To make a fair experimental comparison, we set all hyperparameters using the same strategy. The number of trees in ensemble models (including boosting and bagging) is 200, the sub-sample ratios of instance and feature are 0.8 and 0.6 respectively, and the learning rate is 0.3. The maximum depth of each tree in forest-based (GRF, DML-RF and DR-RF) and boosting-based (meta learners and GBCT) methods is 10 and 3, respectively, where it should be noted that due to the respective characteristics

of the bagging and boosting frameworks, the trees in random forests are generally deeper. Meta learners employ LightGBM as base models and causal forests utilize sub-sampled honest random forest (Athey and Imbens, 2016).

## 4.1 Simulation Studies

**Data Generating Process.** Here we shall give a conceptual introduction to our data generating processes, whose implementation details can be found in Supplementary Materials. We generate a $p$-dimensional confounder vector $X$ from $\sum_{s=1}^{S} \mathrm{I}(G = s)\mathcal{N}(\mu_s, \Sigma_s)$, where $G \in \{1, 2, \cdots, S\}$ is a hidden variable following a predetermined discrete distribution, and indicates which latent cluster each subject belongs to. The treatment-independent feature vector $W$ of $p$-dimension is developed similarly. We then generate the potential outcomes by the following factor augmented auto regressive process of order one, for each $1 \le k \le m$,

$$Y_{t_k}^{(0)} = \rho Y_{t_{k-1}}^{(0)} + (1 - \rho)\{\alpha(X, W) + f(X, W, \lambda_t) + \varepsilon\}, \quad Y_{t_m}^{(1)} = Y_{t_m}^{(0)} + g(X) + \varepsilon_1,$$

where $\rho \in [0, 1]$ is the individual-specific auto-correlation coefficient following a truncated normal distribution $\bar{\mathcal{N}}(\mu_\rho, \sigma_\rho^2; [0, 1])$, $\lambda_t$ is the time-varying factor, the initial value $Y_{t_0}^{(0)} = \beta(X, W)$, $\{\alpha(\cdot), \beta(\cdot), f(\cdot), g(\cdot)\}$ are specified functions described in the supplementary material, and $\varepsilon, \varepsilon_1$ are mutually independent noise items subject to $\mathcal{N}(0, 1)$. To ensure the samples are drawn from the stationary distribution of the data generating process, we use outcomes with time larger than a threshold $t_T$ as observations.

We assign treatment for each unit according to its latent cluster, such that $P(D = 1 \mid G = s) - P(D = 0 \mid G = s) = (-1)^s \phi$ for $s = 1, 2, \cdots, S$, where $\phi \in [0, 1]$ is a specified value. The parameter $\phi$ measures the degree of balance between the treated and control populations; as $\phi$ rises from zero to one, the imbalance gradually increases. In particular, $\phi = 0$ means the treated group has the same population as the control group (i.e., randomized controlled trial), whereas $\phi = 1$ means all treated group's units are from different latent clusters than the control group's. In following experiments, we set $S = 2$, $\mu_\rho = 0.7$, $\sigma_\rho = 0.2$, and the dimension of feature space $2p = 20$. A total of 20000 samples are generated and randomly split into training and validation sets 10 times.

**No Unmeasured Confounding Scenario.** The confounding vector $X$ blocks the backdoor path from treatment to potential outcomes. Thus, given $X$, potential outcomes $Y_{t_m}^{(d)}$ is independent of $D$ for $d \in \{0, 1\}$. Suppose that we observe all confounders $X$, and apply GBCT as well as benchmarks to estimate the conditional treatment effects. We compare our proposed method with benchmark algorithms on the mean absolute error (MAE) of conditional average treatment effects, i.e., $\mathrm{MAE}_{\mathrm{CATE}} = n^{-1} \sum_{i=1}^{n} |\hat{\eta}(X_i) - \eta_i|$, where $\eta_i$ is the true value of the $i$th individual's treatment effect. Table 1 shows the results. As we can see, GBCT outperforms the benchmark algorithms in most cases, especially when $\phi$ is larger, suggesting that it is beneficial to reduce the empirical confounding bias in case of no unmeasured confounding. As $\phi$ increases towards one, except for GBCT and GRF, other methods generally lead to severe bias and significant performance degradation due to low overlap. Moreover, we perform ablation experiments to compare GBCT with its two variants: the variant GBCT-ND without using the confounding entropy $\widehat{\mathrm{H}}(\boldsymbol{Q})$, and the other variant GBCT-B using solely the first term $\widehat{\boldsymbol{b}}_t^2(Q_j)$ in (9) as $\widehat{\mathrm{H}}_t(Q_j)$ to construct the de-bias loss. The results presented in Table 1 demonstrate the important role of our proposed confounding entropy in making the estimates less sensitive to the degree of imbalance between the treated and control populations.

**Unmeasured Confounding Scenario.** We consider the scenario where the confounding vector $X$ are not fully observed. For estimation and evaluation, we retain only the last five dimension of the confounding vector and discard the rest. When $\phi = 0$, the treatment is assigned randomly, resulting in no confounding bias. However, as $\phi$ approaches one, the impact of confounding on treatment assignment grows stronger. Therefore, a large $\phi$ suggests a substantial confounding bias due to unmeasured confounding. In this case, most traditional methods may fail to adjust for bias without further assumptions, while in contrast, GBCT can approximate CATT well. We use $\mathrm{MAE}_{\mathrm{CATT}} = (\sum_{i=1}^{n} D_i)^{-1} \sum_{i=1}^{n} D_i |\hat{\eta}(X_i) - \eta_i|$ as the criterion to evaluate the performance of algorithms, and report the results in Table 1. As expected, GBCT yields smaller mean absolute errors than other methods, and the gap becomes more pronounced as $\phi$ increases, indicating that GBCT can offer a more robust and compelling solution for estimating conditional treatment effects in the

Table 1: MAE (mean±s.d.) of each algorithm on the simulated data. Scenario I represents the absence of unmeasured confounding, whereas II represents the presence of unmeasured confounding. In addition, ●/○ indicates whether GBCT is statistically superior/inferior to the benchmark algorithms (pairwise t-test at 0.05 significance level).

| | $\phi$ | meta learners Slearner | Tlearner | Xlearner | causal forests DR-RF | DML-RF | GRF | GBCT | GBCT-ND | GBCT-B |
|---|---|---|---|---|---|---|---|---|---|---|
| I | 0.2 | 1.26 ± 0.09● | 1.16 ± 0.07● | 1.06 ± 0.05 | 1.23 ± 0.01● | 1.28 ± 0.01● | 1.26 ± 0.01● | 1.05 ± 0.01 | 1.10 ± 0.01● | 1.06 ± 0.01 |
| | 0.5 | 1.46 ± 0.13● | 1.50 ± 0.17● | 1.19 ± 0.10 | 1.29 ± 0.02● | 1.34 ± 0.03● | 1.31 ± 0.02● | 1.25 ± 0.03 | 1.30 ± 0.02● | 1.34 ± 0.04● |
| | 0.8 | 1.52 ± 0.11● | 2.02 ± 0.46● | 1.75 ± 0.53● | 1.51 ± 0.23● | 1.48 ± 0.08● | 1.39 ± 0.04● | 1.31 ± 0.02 | 1.41 ± 0.04● | 1.45 ± 0.04● |
| | 0.9 | 1.48 ± 0.03● | 1.84 ± 0.22● | 1.43 ± 0.21● | 1.80 ± 0.44● | 1.44 ± 0.08● | 1.41 ± 0.12● | 1.28 ± 0.03 | 1.44 ± 0.06● | 1.43 ± 0.10● |
| | 1 | 4.56 ± 0.13● | 3.28 ± 1.59● | 2.46 ± 0.99● | 4.73 ± 0.08● | 16.02 ± 9.39● | 1.96 ± 0.20● | 1.71 ± 0.06 | 3.89 ± 0.57● | 2.85 ± 0.34● |
| II | 0.2 | 1.48 ± 0.07● | 1.97 ± 0.53● | 1.54 ± 0.28● | 1.60 ± 0.02● | 1.66 ± 0.02● | 1.28 ± 0.01● | 1.07 ± 0.01 | 1.11 ± 0.01● | 1.08 ± 0.02 |
| | 0.5 | 1.59 ± 0.08● | 3.54 ± 1.90● | 2.81 ± 1.75● | 1.70 ± 0.09● | 1.72 ± 0.05● | 1.31 ± 0.02● | 1.25 ± 0.02 | 1.29 ± 0.02● | 1.33 ± 0.04● |
| | 0.8 | 1.72 ± 0.09● | 4.10 ± 1.72● | 4.07 ± 2.07● | 3.94 ± 3.31● | 2.14 ± 0.43● | 1.40 ± 0.06● | 1.29 ± 0.04 | 1.34 ± 0.04● | 1.52 ± 0.06● |
| | 0.9 | 2.30 ± 0.07● | 2.63 ± 0.29● | 2.35 ± 0.23● | 2.13 ± 0.03● | 3.10 ± 0.12● | 1.72 ± 0.03● | 1.38 ± 0.09 | 1.92 ± 0.04● | 1.45 ± 0.08● |
| | 1 | 3.84 ± 0.24● | 6.32 ± 3.49● | 4.64 ± 2.94● | 5.20 ± 0.06● | 2.84 ± 0.48● | 2.38 ± 0.17● | 1.81 ± 0.10 | 3.19 ± 0.75● | 2.65 ± 0.76● |

presence of unmeasured confounding. We also compare the performance of GBCT with GRF at the single-tree scale on removing confounding bias, showing that our method can be more effective, see the Appendix. In the ablation experiment, the results confirm the effectiveness of our proposed confounding entropy for reducing bias, especially in the presence of strong unmeasured confounding.

Table 2: Description of the credit card balance dataset. The biased observational dataset only consists of samples among the underlined instances. The features consist of the time-invariant covariates and the balance of credit card account over the last eight months.

| Risk | Number of Instances 0 | 2k | 3k | 6k | Number of Features |
|---|---|---|---|---|---|
| medium-risk | 389477 | 390928 | 391211 | 391215 | 87+8 |
| low-risk | 456773 | 459762 | 459443 | 460352 | |

Table 3: The mean absolute error $\text{MAE}_{\text{ATE}}$ (mean± s.d.) of each algorithm on the credit card balance dataset. The left and right sides are trained on the biased observational and the trial data, respectively. We treat the users with no increase in credit line as the control group.

| | risk | Biased 2k | 3k | 6k | RCT 2k | 3k | 6k |
|---|---|---|---|---|---|---|---|
| TLearner | medium-risk | 49.1 ± 39.8 | 106.5 ± 119.7 | 178.6 ± 116.4 | 92.6 ± 75.3 | 75.2 ± 55.6 | 76.9 ± 39.1 |
| | low-risk | 57.0 ± 39.0 | 104.8 ± 122.2 | 197.7 ± 87.4 | 87.8 ± 76.9 | 51.8 ± 55.4 | 53.9 ± 34.7 |
| XLearner | medium-risk | 88.7 ± 129.0 | 85.0 ± 139.3 | 99.6 ± 132.9 | 98.8 ± 106.4 | 93.0 ± 81.2 | 93.0 ± 48.7 |
| | low-risk | 62.4 ± 129.0 | 99.4 ± 139.3 | 127.1 ± 132.9 | 97.0 ± 106.4 | 81.5 ± 81.2 | 98.1 ± 48.7 |
| SLearner | medium-risk | 207.3 ± 26.3 | 253.4 ± 37.8 | 261.9 ± 59.4 | 12.9 ± 18.8 | 23.4 ± 9.7 | 24.7 ± 10.9 |
| | low-risk | 95.1 ± 28.4 | 100.4 ± 33.7 | 45.1 ± 39.6 | 41.3 ± 8.4 | 59.0 ± 29.2 | 100.9 ± 34.9 |
| DR-RF | medium-risk | 249.6 ± 9.3 | 317.7 ± 12.2 | 408.6 ± 9.4 | 8.6 ± 7.5 | 14.2 ± 7.73 | 48.4 ± 8.8 |
| | low-risk | 149.0 ± 6.5 | 162.8 ± 8.9 | 151.2 ± 11.1 | 35.0 ± 5.0 | 31.7 ± 6.9 | 55.5 ± 9.4 |
| DML-RF | medium-risk | > 999.9 | > 999.9 | > 999.9 | **8.5 ± 6.7** | 15.7 ± 8.6 | 56.4 ± 8.6 |
| | low-risk | > 999.9 | > 999.9 | > 999.9 | 34.5 ± 4.9 | 32.8 ± 6.4 | 62.4 ± 9.3 |
| GBCT | medium-risk | **10.6 ± 6.1** | **27.5 ± 13.6** | **61.5 ± 29.5** | 13.2 ± 7.6 | **8.0 ± 6.0** | **14.2 ± 9.2** |
| | low-risk | **8.4 ± 5.8** | **11.5 ± 8.6** | **19.3 ± 22.5** | 21.7 ± 5.2 | **11.6 ± 7.4** | **22.8 ± 9.9** |

## 4.2 Real Data: Credit Card Balance Dataset

The dataset comes from a randomized controlled trial (RCT) by a commercial finance company aimed at assessing users' heterogeneous responses to increasing credit line of credit card[3]. The trial employs a stratified random assignment design with strata based on risk, dividing users into low-risk and medium-risk. Within each stratum, users are randomly assigned to one of four treatment groups:

---

[3] 1. The dataset does not contain any Personal Identifiable Information (PII). 2. The dataset is desensitized and encrypted. 3. Adequate data protection was carried out during the experiment to prevent the risk of data copy leakage, and the dataset was destroyed after the experiment. 4. The dataset is only used for academic research, it does not represent any real business situation.

increasing credit line by 0, 2000, 3000 or 6000 converted to some currency. In this study, the outcome of interest is the balance of the credit card account in the following month after treatment, i.e., the total amount of debt owed to the credit card. In addition, we observe 87-dimensional covariates at the individual level, as well as outcome values over the last eight months.

In real-world financial scenarios, however, increasing credit line always favors low-risk users; such preference can lead to confounding bias in estimation. In order to assess the performance of different methods in the real-world scenario, we artificially construct biased observational data by retaining only medium-risk users with no credit line increase and low-risk users with credit line increase. See Table 2 for details of the trial dataset and the biased observational dataset.

We are interested in the causal effects of increasing credit line on balance. Unlike simulated data, we do not know all potential outcomes for each user, so we cannot use $\text{MAE}_{\text{CATE}}$ for evaluation. Fortunately, within each risk stratum, due to the randomness of treatment allocation, the ground-truth of the average treatment effect is approximately the mean difference between any two treatment groups in the trial dataset, denoted as $\eta_{\text{ATE}}$. Note that in RCT, the average treatment effect is equal to the average treatment effect on the treated group. Therefore, we compare our proposed method GBCT with the benchmarks on the mean absolute error of the average treatment effect, $\text{MAE}_{\text{ATE}} = \left| n^{-1} \sum_{i=1}^{n} \hat{\eta}(X_i) - \eta_{\text{ATE}} \right|$. To assess variability, we randomly split the entire samples into two folds and repeat this step ten times, each time using one fold as unbiased test data and the other to construct the training data. Given the estimates obtained from the training data, mean absolute errors are calculated separately in each risk stratum of the test data.

The results are reported in Table 3. It is impressive to observe that on both trial and biased data, GBCT significantly outperforms the benchmarks in terms of mean error and standard deviation. Although the benchmarks DR-RF and DML-RF do not perform poorly on the trial data, they suffer severely from bias regarding the observational data; a possible reason is the low overlap between the low-risk and medium-risk strata. As a comparison, GBCT suffers substantially less from bias than others. These findings are in line with those of the prior simulation studies. Therefore, we recommend using GBCT in practice to gain a more stable and robust estimate of causal effects.

## 5    Discussion

In this work, we propose a simple yet powerful method, debiased causal tree, to remove confounding bias caused by unmeasured confounding in the estimation of causal effects (e.g., CATT). Our method only hinges on the availability and transportability of historical controls, i.e., outcomes of units before the treatment, to ensure CATT to be partially identifiable. This allows our method to be readily applicable in a wide variety of scenarios, especially in clinical trials (Viele et al., 2013). Moreover, our method can be generalized to integrate observational data and external control (Li et al., 2021, 2022), with a much weaker transportability condition than existing work (Shi et al., 2022). Nonetheless, our method suffers from several limitations, e.g., time-invariant confounders assumption and failure to provide principled uncertainty quantification as Bayesian causal forests do (Hahn et al., 2020), which needs future investigation.

## Acknowledgments

This work is supported by Ant Group. We thank the three anonymous referees for their constructive suggestions and thoughtful comments, which helped to improve the quality of the paper.

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
