# OpenReview forum: "Debiased Causal Tree: Heterogeneous Treatment Effects Estimation with Unmeasured Confounding"
_NeurIPS.cc/2022/Conference — NeurIPS 2022 Accept_

### Official Review · Reviewer_ZPaK · 2022-06-20

**Rating:** 6
**Confidence:** 4
**Soundness:** 2 fair
**Presentation:** 3 good
**Contribution:** 2 fair

**Summary:**

The paper considers heterogeneous treatment effect estimation when you have longitudinal observations of outcomes pre-treatment. The main contributions are (i) a partial identification result of the CATE in this setting, (ii) a new objective function for finding optimal tree splits, and (iii) a boosting procedure that appears to work well on simulated and real-world data.
The tree splitting criteria seeks a recursive partition of the space of measured confounders that minimizes the sum of two terms. The first term measures the error in fitting the observed post-treatment outcomes using piece-wise constant step functions (i.e. a regression tree). The second term aims to regularize towards partitions for which the distributions of outcomes for treated and control subjects in the pre-treatment period are thought to be small. Under a technical assumptions about how similar confounding bias in the pre-treatment period is to confounding bias in the post-treatment regime, the authors show that they can partially identify the CATE in the potential presence of unmeasured confounding.

===

After the response period, I have raised my score to 6.

**Questions:**

* Can you prove hat regularizing with \hat{H} instead of the sum of squared biases actually results in estimators with smaller variance?If not, you should provide, at a minimum, more justification for not using the sum of squared biases (i.e. the righthand side of Equation 4) as the regularizer in Equation 12. Better would be an empirical demonstration that \hat{H} yielded better point estimates or better uncertainty quantification than using the sum of squared biases

* Relatedly, why did you try to minimize the symmetric cross-entropy between $P_{t,1,j}$ and $P_{t,0,j}$? There are several other choices of distributional differences/distances/divergences. Does closeness in this distance imply closeness in moments?

* To what extent does your framework allow treatment assignment to depend on the potential outcomes in the pre-treatment period? One can readily imagine scenarios where the decision to treat is based on past history. It would be useful to clarify explicitly whether you need to assume treatment is assigned independently of past outcomes.

* Relatedly, what specific causal assumptions do you need to make in order to establish Theorem 1? Presumably you are assuming some sort of positivity assumption and SUTVA. You likely do not assume strong ignorability (i.e.$ (Y(1), Y(0))$ and $D$ are conditionally independent given $X$) and it may be helpful to point this out.

* How does one set $\lambda$ in practice (generally) and how did you set it in your experiments? Is it just through cross-validation?

* Does your method provide any uncertainty quantification? As presented, I don't think it can easily. It may be useful to compare the proposed method to Bayesian Causal Forests, which does provide a type of uncertainty quantification while still flexibly targeting the CATE.



**Limitations:**

The authors do not address any negative societal impacts nor can I easily imagine any for this particular work.
The authors do not explicitly mention any methodological or theoretical limitations. I would encourage them to elaborate on some ways to extend their results in the discussion, e.g., by discussing potential relaxations of specific assumptions or implementation choices.

**Strengths And Weaknesses:**

Strengths: I commend the authors for an original approach to the difficult problem of estimating the CATE in the potential presence of unmeasured confounding. On the whole, I thought the paper was well-written and reasonably clear. I think that the proposed method has the potential to be quite useful to practitioners.

That said, my enthusiasm for the paper is tempered by several concerns about specific modeling and implementation choices and the corresponding assumptions. I will summarize these concerns here and expand on them in the Questions section.

* I am somewhat concerned about the transition from Equation 8 to Equation 9. As written, the assertion in line 159 about the log density of $P_{t,1,j}$ being $-(y_t - \theta^{*(d)}_{j})^{2}$ is true only when $y_t$ is normally distributed with unit variance. You seem to implicitly assume that no matter the partition $Q_j$ or value of $D,$ the outcomes are normally distributed with known and fixed variance. I find the fixed & known variance assumption somewhat unrealistic.

* Additionally, since $\theta_{t,j}^{*(d)}$ is typically unknown, you plug in the sample mean in Equation 8. This may be reasonable when $Q_{j}$ contains a large number of treated and control observations. However, it seems much less reasonable when there are not many of either.  As such the expression in Equation 9 may not be a particularly accurate estimate of the true symmetric cross-entropy between $P_{t,1,j}$ and $P_{t,0,j}.$ I'd go further and say that the expression in Equation 9 represents the symmetric cross-entropy between $P_{t,1,j}$ and $P_{t,0,j}$ under extremely restrictive conditions, namely that the $y_{t,j}$'s are centered around the corresponding sample mean with variance 1 for all choices of $Q_{j}.$

* I found the motivation for regularizing based on historical cross-entropy loss somewhat lacking. The identification result suggests that one should strive to make $\sum_{k = 1}^{m-1}{\sum_{j = 1}^{q}{b_{t_{k}}^{2}(Q_{j}^{**})}}$ as small as possible. You write in lines 150-151 that minimizing this expression can "give rise to partitions that yield estimators with large variance." It is not at all clear that your suggested regularizer avoids this pathology.

---

> ### Author Response · Authors · 2022-08-02
> **Response to Reviewer ZPaK**
>
> We greatly appreciate your positive feedback and valuable comments. We have carefully addressed all the points you have raised and improved the paper accordingly, as detailed in the point-by-point response below. We first reply to the your concerns of our weakness that are itemized with W#, and then respond to your questions one by one that are itemized with Q#. A latex-generating PDF version can be found in the Supplementary materials.
>
> - W1. Thanks for pointing out issues in our paper that may cause misunderstanding. We do *not* require that log-density of $Y_t^{(0)}\mid D=1, X\in Q_j $ and $Y_t^{(0)}\mid D=0, X\in Q_j$ are exactly normally distributed, but use Gaussian distribution to approximate them.  Specifying $P_{t,d,j}$ as a normal distribution $N($ $\theta_{t,j}^{\*(d)}$, $\sigma_{t,j}^{\*(d)})$ is to only consider the first and second moment of $Y_t^{(0)}\mid D=d, X\in Q_j$. In this case, the symmetric cross entropy $\widehat H_{j,t}$ between  $N($ $\theta_{t,j}^{\*(0)}$, $\sigma_{t,j}^{\*(0)})$ and  $N($ $\theta_{t,j}^{\*(1)}$, $\sigma_{t,j}^{\*(1)})$ can be estimated by
>  $$
>      \widehat H_{j,t}=- \log(2\pi)+ \log(\widehat \sigma_1)+\log(\widehat \sigma_0)+ \frac{\widehat \sigma_1^2+(\widehat \theta_{t,j}^{(0)} - \widehat \theta_{t,j}^{(1)} )^2}{\widehat \sigma_0^2}+\frac{\widehat \sigma_0^2+(\widehat \theta_{t,j}^{(0)} - \widehat \theta_{t,j}^{(1)} )^2}{\widehat \sigma_1^2},
> $$
>  where $\widehat \sigma_1,\widehat \sigma_0$ denote sample variances and $\widehat \theta_{t,j}^{(1)}, \widehat \theta_{t,j}^{(0)}$ denote sample mean of $Y_t^{(0)}\mid D=1,X\in Q_j$ and $Y_t^{(0)}\mid D=0,X\in Q_j$, respectively. Noting that variances $\widehat \sigma_1,\widehat \sigma_0$ are denominators, $\widehat H_{j,t}$ is very sensitive to magnitude of $\widehat \sigma_1$ and $\widehat \sigma_0$. More specifically, $\widehat \theta_{t,j}^{(0)} - \widehat \theta_{t,j}^{(1)}$ is unlikely to be penalized towards zero when $\widehat \sigma_1$ and $\widehat \sigma_0$ are large, which may give rise to a $Q_j$ where $Y_t^{(0)}\mid D=d,X\in Q_j\ (d=0,1)$ are very noisy (likely to happen when sample size in $Q_j$ is small) and $\widehat \theta_{t,j}^{(0)} - \widehat \theta_{t,j}^{(1)}$ is relatively large. In order to emphasize the role played by $\widehat \theta_{t,j}^{(0)} - \widehat \theta_{t,j}^{(1)}$ in finding the partition, we simply set the denominators as a constant, which coincidentally equals to the empirical symmetric cross entropy between $N($ $\theta_{t,j}^{\*(0)}$ $,1)$ and $N($ $\theta_{t,j}^{\*(1)}$ $,1)$.
>  Therefore, a fixed and known variance assumption is not imposed but serves as an approximation of true distributions of $Y_t^{(0)}\mid D=d,X\in Q_j\ d=0,1$. Empirically, this approximation works well; see Table 1 in our revised article for detailed simulation results.
>
> - W2. Thanks for your comments. Our explanation for the issue that ``$y_{t,j}$'s are centered around the corresponding sample mean with variance 1 for all choices of $Q_j$'' can be found in the response to your first concern. We agree with you that when sample size in $Q_j$ is small, plug-in estimation of the symmetric cross entropy between $P_{t,1,j}$ and $P_{t,0,j}$ can be  problematic. However, applying some techniques such as pruning, our single debiased causal tree is not very deep such that sample size in each $Q_j$ is not very small, and thus accuracy of estimating confounding entropy is acceptable.  Pruning technique is also used in constructing causal trees proposed by Athey and Imbens (2016, Proc. Natl. Acad. Sci.; 113(27):7353--7360).
>  The price to pay of maintaining a large sample size in $Q_j$ is the risk of underfitting which can introduce a non-vanishing bias. To eliminate the underfitting bias, we further combine multiple debiased causal trees through gradient boosting.
>
> - W3. Thanks for your comments. This comment is closely related to your first question, and responses can be found therein.

---

> > ### Author Response · Authors · 2022-08-02
> > **Cont'd Response to Reviewer ZPaK**
> >
> > - Q1. Thank you for your insightful comment and suggestion. Equation 9 of our article says that $\widehat{H}_t(Q_j)$ can be decomposed into a sum of squared biases and sample variances of local estimators, which suggests that, compared to only squared biases, regularizing with $\widehat{H}_t(Q_j)$ leads to estimators with smaller variances. In general, as we reply to your second question, symmetric cross entropy of $P,Q$ with density function being $p,q$ can be reformulated as
> > $\mathrm{H}(P,Q)=\mathrm{Entropy}(P)+\mathrm{Entropy}(Q)+\mathrm{KL}(Q,P)+\mathrm{KL}(P,Q)$; therefore, regularizing with confounding entropy does have the ability to control model complexity.
> > As you suggest, we demonstrate empirically the gain of regularizing with $\widehat{\mathrm{H}}$ against the sum of squared biases in Table 1 of the new vision; we also present the results here, see Table R.1 in a more formal version of response in supplimentary materials.
> > Our experimental results confirm the superiority of regularizing with confounding entropy over squared biases in terms of estimation error, and the superiority is more significant as the level of unmeasured confounding and imbalance of treatments (in the absence of unmeasured confounding) increases, which supports our analysis.
> >
> > - Q2. Thanks for your question. Conventionally, effects introduced by unmeasured confounding at time $t$ are characterized by $b_t=E(Y_t^{(0)}\mid D=1, X) - E(Y_t^{(0)}\mid D=0, X)$, but it only accounts for differences at the first moment. We propose a general framework to model differences between the conditional outcome distribution of the treated and control groups, aiming to adjust for high-order variations introduced by unmeasured confounders that $b_t$ cannot.
> > Based on this motivation, symmetric cross entropy serves as a well-tested choice, but is *not statistically necessary*.
> > As pointed by you and reviewer J9bc, in principle we can use other alternatives such as integral probability metric (IPM) and Csisz\'{a}r's $f$-divergence to measure the discrepancy between distributions. We highlight that symmetric cross entropy enjoys several advantages.
> > In fact, symmetric cross entropy is closely related to Kullback--Leibler (KL) divergence that is an $f$-divergence with $f(t)=t\log t$; more specifically, the symmetric cross entropy of $P,Q$ with density function being $p,q$ is
> > \begin{aligned}
> >     \mathrm{H}(P,Q)&=-E_P[\log q(X)] - E_Q[\log p(X)]\\\\
> >     &=-E_P[\log p(X)] - E_{Q}[\log q(X)]+E_{P}[\{q(X)/p(X)\}\log \{q(X)/p(X)\}]\\\\
> >     &\qquad\qquad+ E_{Q}[\{p(X)/q(X)\}\log \{p(X)/q(X)\}]\\\\
> >     &=\mathrm{Entropy}(P)+\mathrm{Entropy}(Q)+\mathrm{KL}(Q,P)+\mathrm{KL}(P,Q).
> > \end{aligned}
> > Noting that Shannon's entropy measures variability of random variables, this reformulation of symmetric cross entropy raises an intriguing interpretation of our proposed confounding entropy: regularizing with confounding entropy can *not only* reduce differences between the conditional outcome distribution of the treated and control groups, *but also* control their model complexity and thus avoid overfitting. Also, this observation suggests that closeness in symmetric cross entropy does not imply closeness in moments. Indeed, the maximum mean discrepancy (MMD) $d_K(\cdot,\cdot)$ with kernel function being $K(y_1,y_2)=\phi(y_1)^T\phi(y_2)$ $\phi(y)=(y,y^2)^T, y\in\mathbb{R}$ measures the closeness in moments since $d_K^2(P,Q)=(E_P[X]-E_Q[X])^2+(E_P[X^2]-E_Q[X^2])^2$ for any $P$ and $Q$.
> > Another reason why we choose symmetric cross-entropy, is that we want to accommodate more types of data in a unified notion. For example, when observed outcomes are continuous random variables, we can approximate $P_{t,d,j}$ with Gaussian distributions, and for binary outcomes, we can approximate $P_{t,d,j}$ with Bernoulli distributions.
> > Surely, extension of confounding entropy to a more general class of metrics is possible; however, several issues exist in choosing function class when using  IPM and $f(\cdot)$ when using Csisz\'{a}r's $f$-divergence, e.g., overfitting due to too complex function class which may take noises as signals, misspecification raised by an unsuitable $f(\cdot)$, and potentially expensive time cost for computing these alternatives. These issues can be left for future work.

---

> > > ### Author Response · Authors · 2022-08-02
> > > **Cont'd Response to Reviewer ZPaK**
> > >
> > > - Q3. Thanks for you insightful question. In principle, our method allows dependency between historical outcomes and treatment assignment, but to highlight our main idea, we implicitly exclude this situation and assume that treatment assignment is independent of historical outcomes given covariates and unmeasured confounders (allowed to be time-variant); see Assumption 1 in our revised paper.
> > > Consider the directed acyclic graph in Figure R.2 where past outcomes at $t=t_1<t_2$ can directly affect treatment assignment $D$.  (Figure R.2. can be found in the response of a latex-generating PDF version in the supplementary materials.) Imagine the situation where we do not know the fact that $Y_{t_1}\to D$, and naturally we do not include historical controls as our observed covariates. Accordingly, both historical outcomes $Y_{t_1}$ and $U$ serve as unmeasured confounders *in effect*. Our method is still applicable but unable to use information of $Y_{t_1}$, leading to efficiency and power loss. To highlight our main contribution, we decide to implicitly rule
> > > this situation out by imposing the condition that $Y_{t_m}^{(d)} \perp D \mid (X,U)$ for $d\in \{0,1\}$.
> > >
> > > - Q4. We very much appreciate your helpful comments and suggestions. As you suggest, we impose the SUTVA and consistency assumption to formalize the Rubin causal model; see Lines 88--90. In addition to Assumption 1 of our paper at the first version, we impose the condition that for $d\in \{0,1\}$, $P(D=d\mid X)>0$, and $Y_{t_m}^{(d)} \perp\!\!\!\perp D \mid (X,U)$.
> > > Notice that our positivity condition is weaker than the conventionally assumed version $P(D=d\mid X,U)>0$ for $d=0,1$. This is indeed not surprising because we also require the transportability condition that there exists a partition of observable covariate space, and in each local region, confounding effects change mildly across time; thus, $P(D=d\mid X)>0$ is enough for partial identification.
> > >
> > > - Q5. Thanks for your question. In our experiments, we directly set $\lambda=1$,  based on the consideration that  the two terms in Equation 12 are comparable as both of them can be viewed as an empirical risk. We perform sensitivity analysis  of tuning $\lambda$ in  simulation studies, and the results demonstrate that $\lambda=1$ is an acceptable choice in practice. Results can be found in Table R.3, where is in the supplementary materials.   How to tune hyperparameters in the causal inference literature remains an important open problem.
> > > Cross-validation is commonly used to select hyperparameters in prediction tasks, but often faces difficulties in causal tasks; the reason is that the ground truth of  treatment effect for each unit (no matter belongs to the training data or testing data)  can never be  known, since for each unit only one potential outcome can be observed.
> > >
> > > - Q6. Thanks for your question. As you point out, our method cannot provide uncertainty quantification directly, whereas the confidence interval can be constructed by the Bootstrap procedure.
> > > We add this point as one of the limitations of our method in the discussion, and leave it to the future work.

---

> > > > ### Comment · Reviewer_ZPaK · 2022-08-07
> > > > **response**
> > > >
> > > > Thanks for the detailed response, which addressed many of my reservations.

---

> > > > > ### Author Response · Authors · 2022-08-10
> > > > > **To Reveiwer ZPaK**
> > > > >
> > > > > Thanks for your insightful and constructive suggestions which helped a lot to polish this article. We also feel encouraged about your positive and precise comments on the strength of our paper.  We notice that your positive evaluation of our paper is altered by specific modeling as well as implementation choices, and related assumptions in our paper; in fact our approach is essentially not restricted by the specific working models that are exploited in this paper and is very extensible. We would like to articulate the rationale behind our methods, and clarify any issues that may cause confusion.
> > > > >
> > > > > One fundamental issue you are concerned about is our lacking complete assumptions in our paper, especially on whether treatment assignments can be affected by the historical outcome. We follow your suggestions and have revised our paper; we add SUTVA, consistency, positivity conditions, and assume that treatment assignments are independent of current outcomes given covariates and unmeasured confounders (allowed to be time-variant).
> > > > >
> > > > > You also raised concerns about the choice of symmetric cross entropy for regularization instead of the sum of squared confounding biases. We acknowledge that regularization with the sum of squared confounding biases is more intuitive and straightforward; however, advantages of symmetric cross entropy include (1) being able to exploit high-order informations that squared confounding biases cannot, and (2) control model complexity. We run adequate simulations to confirm the superiority of exploiting confounding entropy over squared confounding biases, whose results are listed in Table 1 in our revised paper. The first advantage can be realized by regularizing with other probability metrics such as integral probability metric (IPM). We remark that the idea we propose to measure the discrepancy between the conditional outcome distribution of the treated and control groups is essential, and our method can be applied with any implementary techniques. It is the second advantage that tempts us to use symmetric cross entropy, which can avoid overfitting but IPMs require to carefully choose the function class.
> > > > >
> > > > > Another issue you mention is that it would be very restrictive to specify outcomes Gaussian distributions. In fact, we are not assuming observed outcomes are Gaussian distributed, but are using Gaussian distributions to approximate their true distribution. Using Gaussian distributions to approximate the distribution of a continuous random variable enjoys several advantages; for example,  estimators (mean, variance) are in closed forms which is easy to compute, and, more importantly, are consistent requiring much less sample size compared to other complex distributions, which is more suitable to tree-based methods.   Given certain prior information, our method can be readily extended to other distributions, e.g., Bernoulli distribution for binary outcomes.
> > > > >
> > > > >
> > > > > Thank you for giving this great opportunity for us to explain these points in detail. We greatly value your opinions, and your support means a lot to us. We would appreciate it if you could reconsider updating our ratings, and we look forward to answering any further questions you may have about our response.

---

> ### Author Response · Authors · 2022-08-06
> **To Reviewer ZPaK**
>
> Thank you again for your insightful comments and inspiring suggestions. In previous replies,  we have carefully addressed the points you raised and improved the paper significantly according to your comments, as detailed in the point-by-point responses. In brief, we demonstrate empirically the gain of regularizing with our proposed confounding entropy against the sum of squared biases in our simulation studies. We explained why our idea is not only restricted to the use of confounding entropy,  but also emphasized the extensibility of our proposed strategy in  leveraging  historical controls to remove confounding bias. We hope our answers can dispel your doubts, and we are willing to provide further explanations if there are any questions. We look forward to hearing back from you.

---

### Official Review · Reviewer_J9bc · 2022-07-07

**Rating:** 9
**Confidence:** 4
**Soundness:** 3 good
**Presentation:** 4 excellent
**Contribution:** 4 excellent

**Summary:**

This paper proposes a new notion, confounding entropy, to measure the discrepancy between the treated and controlled groups. Based on this, the authors propose a new method, the debiased causal tree (more specifically, GBCT), to estimate treatment effects.

**Questions:**

1. What are the differences between your work and "Recursive partitioning for heterogeneous causal effects" "Estimation and inference of heterogeneous treatment effects using random forests" "Generalized random forests"?

2. Why is confounding entropy valid? Is it like the individual version of the integral probability metric? Is there any connection?

3. Why do you need to compare GBCT with meta learners? They are just ways to estimate treatment effects, but the base models can be chosen as linear/tree/neural nets. Similarly, why DML-RF? DML is a doubly robust estimator, and the base models for nuisance parameters can be chosen as linear/tree/neural nets, and you choose RF. In my view, you only need to compare all the regression trees and random forests models that are related to causal inference and analyze the source of gain produced by confounding entropy.

4. In Section 4.2, I don't find any source of data. Where is the real-world dataset and code to reproduce your experiments? I hope code/datasets can be released in the rebuttal stage.

5. I am confused about the Biased dataset in Table 3. Is the control group increasing the credit line by 0? Why increasing the credit line for the low-risk group can bring bias into both low-risk and medium-risk groups? How does the RCT experiments conduct? Is it randomly assigning 0/2k/3k/6k to some people, where 0 is control and 2k/3k/6k is treat?

6. In my view, the most important part is to analyze the gain of confounding entropy, which is missing in other tree/forest-based causalML models. You are not developing a new triply robust estimator or a novel meta learner, so I think there is no need to compare with doubly robust estimators or other meta-learners. So, I think 4-5 well-known tree models for causalML are enough as baselines.

**Limitations:**

The authors do not discuss their limitations. But I think more extensions can be done based on this work.

**Strengths And Weaknesses:**

Originality: 5. This paper gives a new insight into causal inference with tree-based methods. The confounding entropy impresses me a lot.

Quality: 5. This paper is well written, especially for the mathematical parts.

Clarity: 4. It would be great if the context can be connected with former works closely. Readers may be very familiar with Athey's works on regression trees and causal forests, but the content in Section 2 is too dense.

Significance: 3.5. I don't think the improvement is as fair and significant as the paper states. I think the improvement should lie in comparing with the tree-based or forest-based causal ML methods.

___________Discussion period_________________

Since the authors have added adequate experiments to support their theoretical contributions. I strongly recommend an acceptance. I encourage other reviewers to have more discussions in case we miss a good paper. I thus raise my score by 1.

---

> ### Author Response · Authors · 2022-08-02
> **Response to Reviewer J9bc**
>
> We greatly appreciate your positive feedback and valuable comments. We have carefully addressed all the points you have raised and improved the paper accordingly, as detailed in the point-by-point response below. We first reply to the your concerns of our weakness that are itemized with W#, and then respond to your questions one by one that are itemized with Q#. A latex-generating PDF version can be found in the Supplementary materials.
>
> - W1. Thanks for your valuable suggestions. We agree that the  comparisons between our method and  former works on causal forests are important, and we compare our work with causal forests in detail in responding your first question. We emphasize these points in Lines 189--190 and 194--201 of our revised version. However, due to limited paper length, we cannot add  all the comparisons in the main text. We decide to include our response into supplementary materials for interesting readers.
> - W2. Thanks for your valuable suggestions. As you suggest, we add more simulation experiments to evaluate gains of regularizing with confounding entropy; see revised Table 1 in our article and our response to your fourth and sixth question. Also, conceptual differences between our method and forest-based causal models are illustrated in responding to your first question.
> - Q1. Thanks for your question. In the following, we use AI16, WA18, and ATW19 to denote papers you mentioned in the same order (Authors' initials + year of publication), and firstly give a brief review. Notably, the tree methods are built on the condition that there is no unmeasured confounding. From a methodological perspective, AI16, WA18, and ATW19 are closely related. WA18 developed causal forests which generates an ensemble of causal trees proposed in AI16; generalized random forests proposed in ATW19 fit any quantity of interest identified as the solution to a set of local moment equations, which is almost equivalent to causal forests in the case of estimating CATE. In addition to unconfoundedness condition, all of the three methods hinge on the so-called ``honest splitting technique'' that when we grow a specific tree, one data point is exclusively used either for splitting or estimation. Thus, they can only eliminate spurious correlations caused by using the same set of samples to simultaneously select model structure (partition of the covariate space) and estimate parameters based on this selected model structure.
> The differences between our work and the three you mentioned are three-fold.
> The most fundamental and significant difference is that our method can deal with unmeasured confounding but they cannot. In the presence of unmeasured confounding, their methods would lead to an erroneous CATT/CATE estimator even when the sample size is large, e.g., Simpson paradox.
> In contrast, our method gives partial identifiability of CATT; specifically, we quantify the distance between CATT and an identifiable function $\eta(x)$ (defined in Equation 3) in Theorem 1, and the distance can be asymptotically negligible under some conditions.
> Moreover, even in the absence of unmeasured confounding where assumptions of AI16, WA18 and ATW19 are satisfied, our method is more robust to imbalance of treatments; see Section 3 for details. This property enables our method to exploit historical controls under a weaker positivity condition on treatment assignment. As pointed by Reviewer hbGq, one can use historical controls and $X$ to estimate current potential controls $Y_{t_m}^{(0)}$, which, in the context of responding to your question, can be done using causal forests. Notably, however, for identifiability of $E\big(Y_{t_m}^{(0)}\mid D=1, X,Y_{t_k},1\le k\le m-1\big)$, a typical condition is $P(D=d\mid X,Y_{t_k},1\le k\le m-1)>0$, which is stronger than our assumption $P(D=d\mid X)>0$, especially when the number of historical timestamps, i.e., $m-1$, is large.    Lastly, WA18 and ATW19 combine multiple causal trees to generate a random forest, aiming at variance reduction, but we choose the boosting framework to improve performance of a single tree. This different preference for ensemble techniques results from a deep methodological distinction. Methods proposed in WA18 and ATW19 need to reduce variances of trees because trees they construct are very deep, i.e., each leaf contains a very small amount of samples. However, our proposed debiased causal trees cannot be very deep to assure confounding entropy to be accurately estimated, and extra bias would be introduced owing to regularization; therefore, we exploit gradient boosting to further reduce bias. See detailed discussion at Lines 195--203 in our revised paper.
> In fact, we compared our method GBCT and generalized random forests (GRF) empirically; see Table 1 in our paper. In both cases (absence/presence of unmeasured confounding), GBCT outperforms GRF with $95\\%$ confidence level, which supports our analysis.

---

> > ### Author Response · Authors · 2022-08-02
> > **Cont'd Response to Reviewer J9bc**
> >
> > - Q2. Thanks for your insightful question. The rationale behind our method is that (1) control outcomes for treated units are observable historically and thus historical confounding effects can be identified, and (2) confounding effects are assumed to change mildly with respect to time (Assumption 2 in our revised article) such that confounding information can be transferred from historical data to current data.
> > Conventionally, effects introduced by unmeasured confounding at time $t$ are characterized by $ b_t=E(Y_t^{(0)}\mid D=1, X) - E(Y_t^{(0)}\mid D=0, X)$; see Equation 1 for details. Noting that $b_t$ only accounts for differences at the first moment, in our work, we exploit symmetric cross entropy  $H_t$ to model differences between $P(Y_t^{(0)}\mid D=1, X)$ and $P(Y_t^{(0)}\mid D=0, X)$, and we expect that $H_t$ can adjust for high-order variations introduced by unmeasured confounders that $b_t$ cannot. To adapt this idea to tree-based models, we obtain averaged $\widehat H_t(Q)$ across $t\in\{t_1,\ldots,t_{m-1}\}$ as a function of partition $Q=(Q_j, j=1,\ldots,q)$ in Equation 10, which is used as a regularization term in Equation 12.  It adjusts for unmeasured confounding by encouraging to find a partition $\widehat Q=(\widehat Q_j, j=1,\ldots,q)$ such that $P(Y_t^{(0)}\mid D=1, X\in \widehat Q_j)\approx P(Y_t^{(0)}\mid D=0, X\in \widehat Q_j)$.
> > We remark that symmetric cross-entropy is _not statistically necessary_ and in principle we can use other alternatives such as integral probability metric (IPM) and Csisz\'{a}r's $f$-divergence to measure the discrepancy between distributions. IPM with respect to a function class $\mathcal{F}$ is defined as $d_{\mathcal{F}}(P,Q)=\sup_{f\in\mathcal{F}}\big|E_P[f(X)]-E_Q[f(X)]\big|$, and Csisz\'{a}r's $f$-divergence is defined as $div_f(P,Q)=E_Q[f((dP/dQ)(X))]$. By definition, symmetric cross entropy is known to be closely related to Kullback--Leibler (KL) divergence that is an $f$-divergence with $f(t)=t\log t$; more specifically, the symmetric cross entropy of $P,Q$ with density function being $p,q$ is
> > \begin{aligned}
> > &\mathrm{H}(P,Q)=E_P[\log p(X)]+ E_{Q}[\log q(X)]+E_{P}[(q(X)/p(X))\log (q(X)/p(X))] \\\\
> > &+ E_{Q}[(p(X)/q(X))\log (p(X)/q(X))] \\\\
> > &=\mathrm{Entropy}(P)+\mathrm{Entropy}(Q)+\mathrm{KL}(Q,P)+\mathrm{KL}(P,Q).
> > \end{aligned}
> > Therefore, our proposed confounding entropy can be viewed as a symmetric version of KL-divergence between localized distributions, i.e., $Y_t^{(0)}\mid D=d,X\in Q_j, d=0,1$. Moreover, this reformulation of symmetric cross entropy raises an intriguing interpretation of our proposed confounding entropy: regularizing with confounding entropy can not only find a partition that reduce differences between the conditional outcome distribution of the treated and control groups, but also control their model complexity and thus avoid overfitting.
> > Confounding entropy does not belong to a family of IPMs. In fact, IPMs and $f$-divergences are intrinsically different; the family of $f$-divergences and the family of IPMs intersect only at the total variation distance (see, Sriperumbudur et al., Electron. J. Statist. 6: 1550-1599 (2012)).
> > Surely, extension of confounding entropy to a more general $f$-divergences and IPMs is possible; however, several issues exist in choosing function class when using  IPMs, and in choosing $f(\cdot)$ when using Csisz\'{a}r's $f$-divergence, e.g., overfitting due to too complex function class which may take noises as signals, misspecification raised by an unsuitable $f(\cdot)$, and potentially expensive time cost for computing these alternatives. These issues can be left for future work.
> >
> > - Q3. Thank you for this insightful comment and suggestion. We agree that it is essential to assess the gain of our proposed confounding entropy and we have added a detailed discussion on this issue, see the answer for question 6 below.
> > When implementing the conventional methods, we include historical outcomes as covariates, just as suggested by the first reviewer for comparison. Therefore,  our comparison with the conventional methods is intended
> > to evaluate the performances of  two different strategies for utilizing historical controls, one through our proposed confounding entropy  and the other by directly including historical outcomes as covariates.
> > The results confirm that our proposed strategy for integrating the information of historical controls can give a much better performance.
> > We have made this point more clear in
> > our new version, see Lines 258--260, and thank you for pointing out this issue that may cause confusion.

---

> > > ### Author Response · Authors · 2022-08-02
> > > **Cont'd Response to Reviewer J9bc**
> > >
> > > - Q4. The real-world dataset involves users' personal and transactions information, and is not publicly available. We have provided the code of generating simulated data in supplementary material to facilitate replication, and the complete code for implementing our proposed method is in the process of approval and will be released as soon as possible.
> > >
> > > - Q5. Thank you for pointing out these issues that may cause confusion. The reall-world dataset contains 3 million observations from
> > > an RCT, which randomly assigns four treatment arms
> > > (increasing credit line by 0, 2000, 3000 or 6000) to users within the low-risk and  medium-risk strata, respectively. We take the units with a credit line increase of 0 as the control group, and a credit line increase of 2000, 3000 or 6000 as three different treated groups.
> > > In real-world financial scenarios, however, the financial companies  always favor low-risk users, and are unwilling to increase credit line for those with relatively high risks. Therefore, we artificially constructed a biased observational dataset by retaining only the control group of the medium-risk stratum and the treated groups of the low-risk stratum. In this case,
> > > the risk status affects both the treatment assignment and the outcome, and  thus serves as a  substantial confounding. In data analysis, the risk status of each individual is blinded for investigators, and we implement our proposed method by including the observed 87-dimensional covariates as well as outcome values over the last eight months.
> > >
> > > - Q6. We very much appreciate your helpful comments and suggestions. We agree that it is essential to assess the impact of our proposed confounding entropy more in-depth.
> > > In the new version, we made a concerted effort to demonstrate the the gain of confounding entropy by adding discussions and simulations, see Lines 298--302, Lines 316--317 and Table 1. For convenience, we also present the numerical results in Table R.1 which is displayed at the latex-generating PDF version of our response (you can find it in the supplementary materials).  The results confirm that our proposed confounding entropy leads to more substantially accurate and stable estimates across different scenarios.

---

> > > > ### Comment · Reviewer_J9bc · 2022-08-03
> > > > **About Q5**
> > > >
> > > > Thank you for your responses. My concerns regarding technical issues have been well addressed.
> > > >
> > > > But for Q5, I am still confused about your experimental setting. How to interpret "we artificially constructed a biased observational dataset by retaining only the control group of the medium-risk stratum and the treated groups of the low-risk stratum."? According to your explanation, I think you conducted RCT for 3m observations, and the biased dataset is created by retaining medium-risk people who receive 0 and low-risk people who receive 2k/3k/6k. Then how does it correspond to your Table 2?
> > > >
> > > > Both the rebuttal and paper descriptions are very confusing. Could you please give more specific detailed explanations?

---

> > > > > ### Author Response · Authors · 2022-08-04
> > > > > **Reply to Q5**
> > > > >
> > > > > Yes, your understanding of how we construct the biased dataset is exactly right.  Thank you for giving us the opportunity to further explain and clarify the process of real-world data analysis. We have added more specific and detailed explanations below about how we obtain  the results in Table 2.
> > > > >
> > > > >
> > > > > - (i). Step 1: data splitting.  We randomly split the entire samples of the original  RCT dataset into training data and  test data by 10 folds.
> > > > >
> > > > > - (ii). Step 2: construction of the biased dataset.  For the training dataset, we construct the corresponding  biased dataset  by retaining medium-risk people who receive 0 and low-risk people who receive 2k/3k/6k.
> > > > >
> > > > > - (iii). Step 3: estimating causal effects. We apply our proposed methods and benchmarks to estimate  CATTs (2k versus 0, 3k versus 0, and 6k versus 0) based on the  biased  dataset and the RCT dataset that are obtained in Step 2, respectively.
> > > > >
> > > > > - (iv). Step 4:  evaluation on the test dataset.  On the test dataset, we assess the performance of the estimated CATTs within each risk stratum (low-risk and medium-risk).
> > > > > We therefore report the results under different risk strata (low and medium risk) and treatment groups (2k, 3k and 6k) separately in Table 2.
> > > > > By the way, the mean and standard deviation are calculated by 10-fold cross-validation.
> > > > >
> > > > >
> > > > >
> > > > > We hope that the  above answers clarify our process of real-world data analysis and so address your concerns.

---

> > > > > > ### Comment · Reviewer_J9bc · 2022-08-04
> > > > > > **About Q5**
> > > > > >
> > > > > > Thanks for your confirmation.
> > > > > >
> > > > > > Let me further re-emphasize my question. For the biased part in Table 2, how does the experiment conduct on the medium-risk/low-risk group? As you said, for the biased dataset, medium-risk people only receive 0. Among these people, none of them is treated. Then what are values reported in 2k 3k 6k? Similarly, for low-risk people, there are no controlled units.

---

> > > > > > > ### Author Response · Authors · 2022-08-04
> > > > > > > **Reply to Q5**
> > > > > > >
> > > > > > > Thanks for your question.  Our analyzed data come from a  stratified RCT with strata based on risk.  Within each stratum, users are randomly assigned to one of four treatment groups.
> > > > > > >
> > > > > > > In the training phrase, we estimate three CATTs (0 versus 2k, 0 versus 4k, and 0 versus 6k) using the whole biased dataset, and the risk status is latent which serves as an unmeasured confounder. We use the ones who recive 0 in the biased dataset as the control group, and the ones who recive 2k/3k/6k in the biased dataset as the treated groups to estimate CATTs.
> > > > > > >
> > > > > > > In the evaluation phrase, we assess the same CATT (e.g, 0 versus 2k) on the test dataset within different risk strata seperately. The reason that we can perform evaluation within each risk stratum is that the test dataset is randomized (directly from the original whole data) without removing any units, and the ground truth within each risk stratum can be calculated.

---

> > > > > > > > ### Comment · Reviewer_J9bc · 2022-08-04
> > > > > > > > **Final check**
> > > > > > > >
> > > > > > > > Thanks for the reply. Let me do the final confirmation.
> > > > > > > >
> > > > > > > > As you said, for the biased dataset, **medium-risk people only receive 0**. "Among these people, none of them is treated". Is this argument right? If so, for example, for the result of T-learner medium-risk 2k, what is the value 49.1 exactly? Is it the error between the estimated value of $E[Y^{2k} | D=0]$ and the true one? But there is no ground truth since for the biased dataset **medium-risk people only receive 0**.
> > > > > > > >
> > > > > > > > Or I will think that your RCT is separately conducted on medium-risk and low-risk. This seems reasonable to me. Then your former replies and the descriptions in the paper would be very confusing. I'd like to suggest you rewrite them carefully.
> > > > > > > >
> > > > > > > > Finally, how many times does each model run for a specific dataset? Since you have the mean \pm sd, what is the "N" (exp times) of the mean and sd?

---

> > > > > > > > > ### Author Response · Authors · 2022-08-05
> > > > > > > > > **Response to Final Check**
> > > > > > > > >
> > > > > > > > > We very much appreciate your insightful comments and valuable suggestions which have improved this paper significantly, and thank you for pointing out the issues that may cause confusion. We  thoroughly revise our paper according to your comment, and would rewrite and update the real-world data section to avoid any possible confusion.
> > > > > > > > >
> > > > > > > > >
> > > > > > > > >
> > > > > > > > > The following two arguments that you want to  confirm are correct: (i).  in the biased (training) dataset, medium-risk people only receive 0 and none of them is treated, and (ii). our trial is a stratified RCT that is separately conducted on medium-risk and low-risk strata.
> > > > > > > > >
> > > > > > > > > The  biased dataset only serves as a training dataset to learn the causal parameters, while it is worth noting that  the test dataset is not biased and remains an RCT which is separately conducted on medium-risk and low-risk. Therefore, the ground truth  is still known in the test dataset and we evaluate the causal parameters learned from the biased dataset on the unbiased test dataset.
> > > > > > > > >
> > > > > > > > >
> > > > > > > > > For the result of T-learner and medium-risk 2k, the value 49.1
> > > > > > > > > is the mean of $\text{MAE}_{\text{ATE}}$, where
> > > > > > > > >
> > > > > > > > >
> > > > > > > > > $\text{MAE}_\text{ATE}=$
> > > > > > > > >
> > > > > > > > > $abs\big($ $|O^M_{te}|^{-1}$ $\sum_{i\in O^M_{te}}$ $\quad \widehat \eta_{2k}(X_i)-\eta_{\text{ATE}} \big)$
> > > > > > > > >
> > > > > > > > > $\hat\eta_{2k}(X_i)$ is the estimated CATT $E(Y^{(2k)}-Y^{(0k)}\mid X,D=2k)$  based on the biased dataset, $O^M_{te}$
> > > > > > > > > is the set of all medium-risk people in the test dataset, and $\eta_{\text{ATE}}$ is the ground truth $E[\eta_{2k}(X)]$ that can be approximated in the medium-risk stratum of unbiased  test data.
> > > > > > > > >
> > > > > > > > > For the mean $\pm$ sd, we use 10-fold cross validation to construct the biased traning and unbiased test data (splitting 3 million samples into 10 folds and each time use one of them as unbiased test data and the others to construct biased training data), and thus the number of experiment times is 10.
> > > > > > > > >
> > > > > > > > > Thank you very much for your patience, encouragement and positive comments, and we look forward to your feedback if we still have any unclear expressions.

---

> > > > > > > > > > ### Comment · Reviewer_J9bc · 2022-08-05
> > > > > > > > > > **Thanks for the reply**
> > > > > > > > > >
> > > > > > > > > > Understood with thanks. Though the experimental setting and conduction remain unperfect, they do not detract from the overall excellence of the whole paper. I would like to maintain the score. Hope the dataset can be open to the public asap.

---

### Official Review · Reviewer_hbGq · 2022-07-10

**Rating:** 5
**Confidence:** 3
**Soundness:** 2 fair
**Presentation:** 2 fair
**Contribution:** 3 good

**Summary:**

In the observational data, this paper consider the case where covariates and the outcome are collected at multiple timestamps, $t \in \{ t_{1}, \ldots, t_{m} \}$, and only implement treatment $D=d, d \in$ {$0,1$} at the last timestamp $t_m$. In this case, the historical controls means that all treatments are $D=0$ at the timestamp $t \in ${$ t_{1}, \ldots, t_{m-1} $}. The treatment sequence is an known sequence, i.e.,{$0,0,\ldots, 0, d$}.

Using observational data and historical controls, this paper proposes a new notion confounding entropy measure the discrepancy between the conditional outcome distribution of the treated and control groups. Then, they use confounding entropy as splitting rule to divide the data into multiple subgroups and estimate  conditional average treatment effect on the treated group.

----
I thank the authors for addressing my comments. I increase my score to 5.

**Questions:**

- Using observational data and historical controls, can we directly and accurately estimate potential control outcomes with a large amount of historical control data? Is the estimation of conditional average treatment effect on the treated group still a challenge?
- Lines 126-127, what is the relationship between Theorem 1 and the identification of instrumental variables?
- Why not use observed outcomes on the treated group when estimating CATT?
- The data generation mechanism, and estimators $\{\alpha(\cdot), \beta(\cdot), f(\cdot), g(\cdot)\}$ are not detailed in this paper.

**Limitations:**

I don't believe that this method proposed in this paper can reduce the bias from unmeasured confounders that are independent with observed confounders. Besides, in practical, time-invariant covariates and unmeasured confounders assumption is a strict condition that is difficult to satisfy.

**Strengths And Weaknesses:**

**Strengths**

This paper considers a intersting setting about historical controls, and proposes a new notion confounding entropy measure the discrepancy between the conditional outcome distribution of the treated and control groups. The computational feasibility and statistical power are evidenced by simulations and credit card balance dataset.

**Weaknesses**

- Using observational data and historical controls, we can directly and accurately estimate potential control outcomes $\hat{Y}^0$ with a large amount of historical control data. The conditional average treatment effect on the treated group can be easily obtained: $CATT = \mathbb{E}[Y|D=1,X] - \mathbb{E}[\hat{Y}^0|D=1,X]$.
- In Lines 98-102, proper stratification [26] can help mitigate confounding effects from observed confounders but can't reduce the bias from unmeasured confounders that are independent with observed confounders.
- In practical, time-invariant covariates and unmeasured confounders assumption is hardly satisfied in a causal scenario.
- In Lines 104-107, without any prior knowledge, neither constant function nor polynomial function  guarantees robust results.
- Checklist in Lines 492-493: The Assumption 1 is not the full set of assumptions. SUTVA, positive/overlap assumption are missing in this paper. The authors should reframe the assumptions required.
- Figure 1 and Figure 2 in Appendix, what data was used to draw this picture? the authors did not give a clear explanation.
- Not Reproducible. The data generation mechanism, and estimators $\{\alpha(\cdot), \beta(\cdot), f(\cdot), g(\cdot)\}$ are not detailed in this paper.

---

> ### Author Response · Authors · 2022-08-02
> **Response to Reviewer hbGq**
>
>
> We greatly appreciate your positive feedback and valuable comments. We
> have carefully addressed all the points  you have raised and improved the paper accordingly, as detailed in the point-by-point response  below.
> We first reply to the your concerns of our weakness that are itemized with W\#, and then respond to your questions one by one that are itemized with Q\#. A latex-generating PDF version can be found in the Supplementary materials.
> - W1
> Thanks for your insightful comment. In the presence of unmeasured confounding, we in fact _cannot_ accurately estimate potential control outcomes $\widehat{Y}^{(0)}$ with observed covariates and historical outcomes. We provide a detailed explanation of why we cannot in the response to your first question, which we hope will dispel your doubts.
> - W2. Thank you for pointing out this issue that may cause confusion, and we  add the sentence "by exploiting historical controls that contain information of
> unmeasured confounding" in the revised version to avoid misleading. We agree with what you said, but it does not conflict with our method. Different from classical matching methods that only adjust for $X$, our method finds a partition of the covariate space (stratification) in which unmeasured confounders $U$ can also be partially adjusted for because (1) control outcomes for treated units can be ``observed" historically and thus historical confounding effects can be identified, and (2) confounding effects are assumed to change mildly with respect to time (Assumption 2 in our revised article) such that confounding information can be transferred from historical data to current data. In other words, the stratification depends not only on observed covariates $X$ but also unmeasured confounders $U$ through historical controls.
>
> - W3. Thanks for your valuable comment. We make some revision and now we allow unmeasured confounders to be time-varying. The time-invariant assumption of covariates is standard in the literature of causal inference, and serves to simplify our formal definition and theoretical analysis of partial identification of CATT. In fact, this assumption is not as restrictive as it seems to be. In spite of measured confounders being time-invariant, their effects on outcomes can be time-varying; see Assumption 2 in our revised article. Besides, measured confounders that are time-invariant, or at least varying mildly with respect to time, are very common, e.g., sex, age, BMI, etc. When samples are collected over a small time span, i.e., $t_m-t_1$ being small, the time-invariant assumption of confounders can be viewed to hold. Relaxation of this assumption is possible by considering functional regressions (e.g., Ju and Salibi&aacute;n-Barrera 2021, arXiv:2109.02989).
> - W4. Thanks for your  comment. Piece-wise constant functions, or step functions, enjoy point-wise universal approximation ability to the space of measurable functions. In fact, trees, as an example of step functions, have shown a great success in various practical applications. Indeed, our partial identification bound takes the issue you raised into consideration; see Theorem 1. The second term of the upper bound, $\max_{1\le j\le q}\sup_{x\in Q_j^{\*\*}}|\eta^{\*}(x)-\eta^{\*}(Q_j^{\*\*})|$, measures the distance between our target function $\eta^*(x)$ and its step function approximator $\eta^*(Q_j^{**})$, which would be large if step functions are not suitable.
> Surely, to obtain reasonable theoretical guarantees, certain smoothness condition, e.g., bounded variation or high-order differentiability, should be imposed on the target function $\eta^*(x)$, but this is not the key issue we aim to solve in this paper.
>
> - W5. Thanks for your helpful comment. As you suggest, we impose the SUTVA and consistency assumption to formalize the Rubin causal model; see Lines 90--91. We also impose the condition that for $d\in \{0,1\}$, $P(D=d\mid X)>0$, and $Y_{t_m}^{(d)} \perp D \mid (X,U)$.
> Notice that our positivity condition is weaker than the conventionally assumed version $P(D=d\mid X,U)>0$ for $d=0,1$. This is indeed not surprising because we also require the transportability condition that there exists a partition of observable covariate space, and in each local region, confounding effects change mildly across time; thus, $P(D=d\mid X)>0$ is enough for partial identification.

---

> ### Author Response · Authors · 2022-08-02
> **Cont'd Response to Reviewer hbGq**
>
> - W6. Thanks for your valuable comments. The data used to draw the pictures in Appendix are generated in the same way as the simulation. We have added more detailed descriptions of figures in the revised version.
> Figure 1 shows the effectiveness of a single tree of GBCT on reducing confounding bias. Let $ \widehat Q_j^{\mathrm{GBCT}} ,j=1,\ldots, q_1$ be the partition of a single tree of GBCT, and let $\widehat Q_j^{\mathrm{GRF}}, j=1,\ldots,q_2$ be that of GRF. Given our generated data, the $y$-axis denotes $\sum_{j=1}^{q_1}$  $| \widehat b_{t_m}$ $(\widehat Q_j^{\mathrm{GBCT}}) |$ and $\sum_{j=1}^{q_2} | \widehat b_{t_m}(\widehat Q_j^{\mathrm{GRF}}) |$, respectively, where
> $$
> \widehat b_{t_m}(Q_j)=\sum_{i=1}^n\frac{ I ( X_i\in Q_j, D_i =1) Y_{i,t_m}^{(0)}}{| \{ i: X_i\in Q_j, D_i=1 \} |}-\frac{ I(X_i\in Q_j, D_i =0) Y_{i,t_m}^{(0)}}{| \{i: X_i\in Q_j, D_i=0\} |},
> $$
> and $Y_{i,t_m}^{(0)}$ is available for $D_i=1$ since it is a simulation. Here, we use an $\ell_1$-type loss because it is the largest norm for vectors such that small confounding bias can be highlighted.
> Figure 2 shows the ablation experiments to demonstrate the gain of the confounding entropy regularization (defined in Equation 10) and the trick of subtracting the empirical pre-treatment confounding bias.  "GBCT-ND" means without confounding entropy regularization $\widehat{\mathrm{H}}(Q)$,  "GBCT-NR" means that the pre-treatment confounding bias is not subtracted when predicting the effect, and  "GBCT-NR-ND" means neither confounding entropy loss nor subtraction of the pre-treatment confounding bias. The $y$-axis means $ \mathrm{MAE_{CATE}}=\frac{1}{n}\sum_{i=1}^n|\hat{\eta}(X_i)-\eta_i|$ defined in Line 293.
> In addition, the performance of "GBCT-ND" has been updated in Table 1 in our revised version.
>
> - W7. Thanks for pointing our missing details. We have provided the code of generating simulated data in supplementary material to facilitate replication, and the complete code for implementing our proposed method is in the process of approval and will be released as soon as possible. Estimators $\alpha(\cdot),\beta(\cdot),f(\cdot),g(\cdot)$ are specified in responding to your fourth question.
>
> - Q1. Thanks for your question. We _cannot_ estimate CATT or potential control outcomes $Y_{t_2}^{(0)}$ by simply adjusting for $X$ and historical controls $Y_{t_1}^{(0)}\ (t_1<t_2)$ *when unmeasured confounders  directly affect both current potential outcomes and treatment assignment.*  As an illustrative example, consider the case where $(U,X)\to Y_{t_1}\to Y_{t_2}, (U,X)\to D, (U,X)\to Y_{t_2} $, where $U$ denotes some unmeasured confounders, $X$ denotes observed covariates, $D$ denotes the treatment, and $Y_{t_1}, Y_{t_2}$ denotes historical and current outcomes, respectively.
> For the ideal case where $Y_{t_2}^{(0)}$ can be accurately estimated, without doubt, CATT can be estimated directly by plugging the estimated $Y_{t_2}^{(0)}$ in the definition (Line 97 in our revised article). However, since $(X,Y_{t_1})$ cannot block the backdoor of $(D,Y_{t_2})$, directly using $(X,Y_{t_1})$ to estimate or predict $Y_{t_2}^{(0)}$ _cannot_ fully adjusted for unmeasured confounders $U$, which would lead to a biased estimate of $Y_{t_2}^{(0)}$ and further an erroneous CATT estimator; e.g., Simpson paradox would occur even when the sample size is large.
> In deep contrast, our method can give partial identifiability of CATT; specifically, we quantify the distance between CATT and an estimable function $\eta(x)$ (defined in Equation 3) in Theorem 1, and the distance can be asymptotically negligible under some conditions.
> Different from classical matching methods that only adjust for $X$, our method finds a partition of the covariate space (stratification) in which unmeasured confounders $U$ can also be partially adjusted for because (1) control outcomes for treated units can be `observed' historically and thus historical confounding effects can be identified, and (2) confounding effects are assumed to change mildly with respect to time (Assumption 2 in our revised article) such that confounding information can be transferred from historical data to current data. These chronologically sampled data do facilitate our identification and estimation of CATT.   Moreover, empirically, we compared several methods such as meta learners and causal forests which exploit historical controls as covariates to estimate CATT with our proposed method GBCT. GBCT yields smaller mean absolute errors than these methods, and the gap becomes more pronounced as  impact of unmeasured confounding increases; see details at Lines 298--302 in our revised article.

---

> > ### Author Response · Authors · 2022-08-02
> > **Cont'd Response to Reviewer hbGq**
> >
> > - Q2. Thanks for pointing out the unclarity in our presentation. We meant to highlight that CATT is partially identifiable (identifiable up to a band) in our case (Theorem 1), and we mentioned instrumental variables as an example that ATE can be partially identified using IV-based methods.
> >
> > - Q3. We _did use_ observed outcomes on the treated group to estimate CATT; see Lines 179--180 and Equation 12 in our revised article.
> > We are concerned that our presentation cannot adequately convey the rationale of our method which leads to your misunderstanding. Our regularized empirical risk minimization, i.e., Equation 12, consists of two parts. The first term measures the distance between our model and observed current outcomes, and the second term is estimated confounding entropy. The second term encourages to find a partition in which $\eta(x)\approx \mathrm{CATT}$ as illustrated in replying your first question. For the first term, we use a tree $T(x;{Q},\widehat \mu^{(1)}( Q))$ to learn $ \sum_{j=1}^q$ $E($ $Y_{t_m}^{(1)}$ $ | D=1, X \in Q_j^{\*})$ $I (x\in Q_j^{\*})$, where we use observed outcomes on the treated group, and another tree  $T(x; Q,\widehat \mu^{(0)}( Q) )$ to learn $\sum_{j=1}^q E(Y^{(1)}_{t_m}\mid D=1, X\in Q_j^*) I (x\in Q_j^*)$. Thus, informally
> >  $$\widehat{\eta}(x) \equiv T(x;\widehat{{Q}},\widehat{{\mu}}^{(1)}(\widehat{{Q}}))-T(x;\widehat{{Q}},\widehat{{\mu}}^{(0)}(\widehat{{Q}}))\approx \eta(x)\approx \mathrm{CATT},$$ where $\widehat{{Q}}$ denotes the solution of Equation 12.
> >
> > - Q4. Thanks for pointing our missing details due to limited length of paper. We provide codes of generating data in supplementary materials. Estimators $\alpha(\cdot),\beta(\cdot),f(\cdot),g(\cdot)$ are specified as follows.
> > The interception function $\alpha(\cdot)$ is
> > $\alpha(X,W)$ $=(\frac{\alpha_1(X\oplus W)}{\sigma_0}$ $+$ $\frac{\alpha_2(X \oplus W)}{\sigma_0}$ $-\frac{\mu_{0}}{\sigma_{0}})$ $\tilde \sigma_{1}$ $+\tilde \mu_{1},$
> > where
> > \begin{align*}
> >      &\alpha_1(Z) = \frac{1}{p_z-4}\sum_{i=0}^{p_z-4} 10\sin(\pi Z_{i} Z_{i+1}) + 20(Z_{i+2} - 0.5)^2 + 10 Z_{i+3} + 5 Z_{i+4},\\\\
> >     &\alpha_2(Z) = \frac{1}{p_z-4}\sum_{i=0}^{p_z-4} \sqrt{Z_{i}^2 + \big[Z_{i+1} Z_{i+2}  - 1 / (Z_{i+1} Z_{i+3})\big]^ 2},\quad  Z\in \mathbb{R}^{p_z},
> > \end{align*}
> > $\mu_0=E\big[\alpha_1(X\oplus W)+\alpha_2(X\oplus W)\big]$ and $\sigma_0^2=\mathrm{var}\big[\alpha_1(X\oplus W)+\alpha_2(X\oplus W)\big]$, $X\oplus W=(X^T,W^T)^T$ represents concatenating two vectors, and $\tilde \mu_1=2$ and $\tilde \sigma_1=2$ are the pre-defined mean and standard deviation.
> > The function $\beta(\cdot)$ is
> > $
> > \beta(X,W)=\theta_b^T(X\oplus W) - E[\theta_b^T(X\oplus W)] $ $+ \tilde \mu_b$,  $\theta_b \sim$ $ N_{2p}(0,I_{2p}),$
> > where $\tilde{\mu}_\beta=2$.
> > The time-varying related function $f(\cdot)$ is
> > $f(W,\lambda_t) = {(\Theta_f^T W)}^T\lambda_t,$
> > where $\Theta_f$ is a $p\times 3$ matrix and  $\lambda_t\in \mathbb{R}^3$ denotes a time-varying factor. In addition, entries of ${\Theta}_f$ are independently sampled from a normal distribution $\mathcal{N}(0, 1)$.
> > The effect function $g(\cdot)$ is
> > ${g}({X})=\frac{\alpha_1({X})/10}{\log(|\alpha_2({X})|+1)}.$

---

> ### Author Response · Authors · 2022-08-06
> **To Reviewer hbGq**
>
> We would like to thank you for the careful review of our work. In the previous replies, a detailed account of the response to your insightful comments and the changes we made to the paper are provided in a point-by-point style. Specifically, from both theoretical and empirical (simulation experiments) perspectives, we have explained why it is not recommended to simply use historical results as covariates to adjust for unmeasured confounding bias. In contrast, we have made a concerted effort to clarify  why our method works under our setting.  In addition, your concerns about our assumptions and replicability are well addressed.
>
> Thank you again for  your  valuable comments which improved this paper significantly, and we have thoroughly revised our paper according to your comments.  We hope our answers dispel your doubts, and we are willing to provide further explanations if there are any questions. We look forward to hearing back from you.

---

> ### Author Response · Authors · 2022-08-10
> **To Reviewer hbGq**
>
> Thanks for your confirmation and insightful comments, which helped to improve this paper.
>
> You raised concerns that our approach may fail if all unmeasured confounders are independent of the observed covariables as well as historical outcomes. We acknowledge that in this special case, historical control data may not provide any useful information; while our focus is on the more usual situations where unmeasured confoundings affect both historical and current outcomes, which is implicit in the identification assumption we propose. Actually, our central assumption for partial identification is quite weak, only requiring that confounding effects caused by unmeasured confounders (can be time-variant) on outcomes change mildly with respect to time. More importantly, our assumption is testable to some extent because the historical confounding effects are observable and thus can be tested for sharp changes. Considering that adjusting for unmeasured confounding bias is a very tough challenge in causal inference, traditional approaches also rely on certain strong assumptions and are not applicable to all situations. For example,  the instrument variable (IV) framework requires no interaction condition, and the proxy variable framework requires the completeness condition. Finally, in the special case where all unmeasured confounders are independent of the covariables as well as historical outcomes, our method generally does not lead to worse results due to the independency.
>
> Another issue you mention is the time invariance of covariates and unmeasured confounders. We have provided a detailed response below, where we make some revisions and allow unmeasured confounders to be time-varying. As we pose treatment arms on the current time, and thus natrually the causal quantity of interest is the treatment effects conditional on the current covariates but not on the historical covariates. In fact, our theorem shows that under our identifiability condition, the elimination of unmeasured confounding bias requires only the help of historical outcomes but not necessarily historical covariates. It is worth noting that this is precisely a positive conclusion and not a negative one! To illustrate, consider a scenario where the historical outcomes are observed but historical covariates are missing, and the historical covariates affect both the treatment assignment and the current outcome directly. In this case, the historical covariates are precisely the unmeasured confounders,  and the estimation based on the current data must be biased. However, our framework shows that only if the effect of historical covariates on historical outcomes changes mildly with respect to time, then the unmeasured confounding bias can be reduced by borrowing strength from historical outcomes. In short, requiring only historical results and not necessarily historical covariates is an important advantage of our framework.
>
> Thank you for giving this great opportunity for us to explain these points in detail. We greatly value your opinions, and your support means a lot to us. We would appreciate it if you could reconsider updating our ratings, and we look forward to answering any further questions you may have about our response.

---

### Author Response · Authors · 2022-08-09
**A brief summary to chairs & all reviewers**

We thank the reviewers for their careful review, constructive comments and positive feedbacks. We have taken the comments and suggestions on board to improve and clarify our work, and we believe the paper has been significantly improved.

How to adjust for unmeasured confounding bias has always been a central issue as well as a tough challenge in causal inference.  One of the main obstacles is that, in general, the identification of causal effects fails to hold without extra information.  To be more specific, with unmeasured confounding, identification does not hold even for fully parametric models; exceptions require fairly restrictive assumptions, e.g., the Heckman selection model. Therefore, for models without imposing parametric constraints,  identification and inference basically require the use of auxiliary data. A traditional approach is using the instrument variable (IV),  and recently some researchers propose to use another kind of auxiliary data called the proxy variable. Both of the two approaches for removing unmeasured confounding bias need additional no interaction or completeness conditions that further limit their use, and such auxiliary variables may be hard to find in practice.

In contrast, we propose to adjust for unmeasured confounding bias by borrowing strength from the historical control data. Fortunately, scenarios with available historical control data are common in fields such as clinical research, finance and sociology, e.g., the patients' past records under conventional medicines. We build the partial identification result by leveraging this kind of auxiliary data. Our central assumption for partial identification is quite weak, only requiring that confounding effects caused by unmeasured confounders (can be time-variant) on outcomes change mildly with respect to time. More importantly, our assumption is testable to some extent, because the historical confounding effects are observable and thus can be tested for sharp changes. Our framework provides a new approach to adjusting for unmeasured confounding bias, especially when the aforementioned auxiliary variables (IV or proxy variables) are not available. In addition, our approach is essentially not restricted by the specific working models that are exploited in this paper and is very extensible.

We believe that our work makes a solid step to establish a framework that allows researchers to leverage historical control data to alleviate unmeasured confounding bias. And we also want to take this opportunity to thank all reviewers again for their effort that they contributed towards reviewing the paper.

---

### Meta-Review · Area_Chair_ucyu · 2022-08-27

**Recommendation:** Accept
**Confidence:** Certain

**Metareview:**

The authors propose a novel way of dealing with unobserved confounding when having access to historical untreated data on the treated subjects and using transportability ideas and finding partitions of the covariate space that minimize the confounding bias using the historical untreated data. They introduce an interesting notion of confounding entropy and use it successfully in practice within a gradient boosted forest framework.

Despite initial reviewer concerns, the authors rebuttal has addressed main concerns.

**Award:**

No

---

### Decision · Program_Chairs · 2022-09-14

Accept